# Transcriptional profiling of Hutchinson-Gilford Progeria syndrome fibroblasts reveals deficits in mesenchymal stem cell commitment to differentiation related to early events in endochondral ossification

**Rebeca San Martin[1], Priyojit Das[2], Jacob T Sanders[1,3], Ashtyn M Hill[1], Rachel Patton McCord[1]***

[1]Department of Biochemistry & Cellular and Molecular Biology, University of Tennessee at Knoxville, Knoxville, United States; [2]UT-ORNL Graduate School of Genome Science and Technology, University of Tennessee at Knoxville, Knoxville, United States; [3]Department of Pathology, University of Texas Southwestern Medical Center, Dallas, United States

**\*For correspondence:**
rmccord@utk.edu

**Competing interest:** The authors declare that no competing interests exist.

**Abstract** The expression of a mutant Lamin A, progerin, in Hutchinson-Gilford Progeria Syndrome leads to alterations in genome architecture, nuclear morphology, epigenetic states, and altered phenotypes in all cells of the mesenchymal lineage. Here, we report a comprehensive analysis of the transcriptional status of patient derived HGPS fibroblasts, including nine cell lines not previously reported, in comparison with age-matched controls, adults, and old adults. We find that Progeria fibroblasts carry abnormal transcriptional signatures, centering around several functional hubs: DNA maintenance and epigenetics, bone development and homeostasis, blood vessel maturation and development, fat deposition and lipid management, and processes related to muscle growth. Stratification of patients by age revealed misregulated expression of genes related to endochondral ossification and chondrogenic commitment in children aged 4–7 years old, where this differentiation program starts in earnest. Hi-C measurements on patient fibroblasts show weakening of genome compartmentalization strength but increases in TAD strength. While the majority of gene misregulation occurs in regions which do not change spatial chromosome organization, some expression changes in key mesenchymal lineage genes coincide with lamin associated domain misregulation and shifts in genome compartmentalization.

## Editor's evaluation

This manuscript is of interest to researchers investigating genetic mechanisms of aging and transcriptional regulation of developmental processes in mesenchyme-derived tissues. In this study, fibroblast cell lines from patients with and without Hutchinson-Gilford Progeria were compared to pinpoint the molecular mechanisms leading to the phenotypes of persons with this condition. The identification of five major dysregulated functional hubs in fibroblast cell lines derived from Hutchinson-Gilford Progeria Syndrome (HGPS) patients provides a unique opportunity for others working on this disorder to utilize animal models to validate the authors' hypotheses.

## Introduction

Hutchinson-Gilford progeria syndrome (HPGS) is a rare disease, characterized by a severe premature aging phenotype (*Gilford and Hutchinson, 1897*; *Hutchinson, 1886*). Patients appear normal at birth, with a disease onset around 2 years of age when they present with slow growth rate, short stature, and marked lipodystrophy with a characteristic loss of subcutaneous fat. With progression, patients experience arthritis, joint contracture, osteoporosis and stiffening of blood vessels. Average life expectancy for HPGS patients is 13.4 years, with myocardial infarction and stroke being the predominant causes of death (*Foundation, 2019*).

HGPS is caused by a thymidine substitution mutation at cytidine 608 (GGC >GGT), within exon 11 of the gene that encodes for Lamin A (LMNA) (*De Sandre-Giovannoli et al., 2003*; *Eriksson et al., 2003*). Lamin A is a structural protein of the nuclear envelope, playing an important role in genome structure and nuclear integrity. The mutation does not result in an amino acid substitution, frame shift or early termination. Rather, it induces the usage of a cryptic splice site and the subsequent deletion of 50 amino acids at the C terminus. In turn, this deletion impedes downstream processing of the protein and results in conservation of C terminus farnesylation (*Davies et al., 2009*). This abnormal protein product is called Progerin. Aggregation of Progerin leads to a disruption of the normal nuclear envelope meshwork, leading to deformed nuclei, nuclear stiffening, and defective mechanotransduction (*Apte et al., 2017*; *Goldman et al., 2004*; *Lammerding et al., 2004*).

Of particular interest, progeria patients present with various defects in all tissues of the mesenchymal lineage. Specifically, regarding bone physiology, patients show reduced stature, generalized osteopenia, thin calvaria, and clavicle regression with absence of medial and lateral ends, as well as resorption of the distal bony phalanges and anterior ribs (*Chawla et al., 2017*; *Cleveland et al., 2012*; *Gordon et al., 2011*; *Nazir et al., 2017*). These phenotypes are more striking when considering that at birth, bone structure appears normal. Previous observations, where fibroblasts from progeria patients showed abnormal levels of aggrecan (*Lemire et al., 2006*) a marker of chondrogenic differentiation, led us to hypothesize that fibroblasts derived from HGPS patients – themselves a cell of the mesenchymal lineage – could harbor vestigial transcriptional signatures to abnormal mesenchymal stem cell commitment, which would become apparent by comparing them with different normal age group controls.

In this study, we conduct a comprehensive analysis of previously published RNA-seq datasets for HPGS fibroblast cells harboring the typical C>T mutation in the LMNA gene and provide transcriptomics data for nine patient fibroblast samples that had no previously reported RNA-seq data. By comparing the transcriptional profile of HPGS fibroblasts (33 datasets, 21 patients, 1–20 years old) with fibroblasts derived from age-matched controls (21 datasets, 16 donors, 1–19 years old), healthy adults, (16 donors, 26–43 years old) and healthy old adults (15 donors, 80–96 years old), we provide insight into important defects in repair biology, metabolism (calcium, lipid), and other areas important to the mesenchymal cell lineage. Our results show that transcription of genes involved in negative regulation of chondrocyte commitment is compromised in fibroblasts from patients that are at the age of onset of postnatal endochondral ossification. We report that some genes central to differentiation commitment into the chondrogenic-osteogenic lineage and identified as misregulated in this study (PTHLH, BMP4, FZD4, and IGF1) show either abnormal genomic architecture or lamin associated domain alterations.

Our results support the hypothesis that defects in the initial steps of chondrogenesis commitment are a potential mechanism for mesenchymal stem cell depletion that later results in abnormal adipogenesis, diminished microvasculature homeostasis, and poor wound repair observed in HGPS patients.

## Results

### Batch correction is essential for comparison among patient and normal cohorts

To determine gene sets that were consistently misregulated in Progeria fibroblasts, we collected both newly generated and previously published RNA-seq data from all available Progeria patient fibroblast cell samples as well as fibroblasts from control individuals in different age groups (*Appendix 1—tables 1–3*; *Fleischer et al., 2018*; *Ikegami et al., 2020*; *Köhler et al., 2020*; *Mateos et al., 2018*).

**Table 1.** Number of up/down-regulated genes per age comparisons.

| Comparisons | Upregulated genes | Downregulated genes |
|---|---|---|
| Young patients – Age matched | 260 | 63 |
| Young patients – Adult control | 574 | 241 |
| Young patients – Old adult control | 984 | 435 |
| Teenaged patients – Age matched | 237 | 81 |
| Teenaged patients – Adult control | 1873 | 1138 |
| Teenaged patients – Old adult control | 1872 | 1022 |
| | | |
| Young Control – Adult Control | 28 | 25 |
| Young Control – Old Control | 817 | 421 |

The online version of this article includes the following source data for table 1:

**Source data 1.** Metascape outputs for gene ontology analysis.

When the resulting transcriptomics data were analyzed by principal component analysis (PCA), datasets clustered according to laboratory of origin regardless of diagnosis, making direct comparisons impossible (*Figure 1—figure supplement 1A*), as has been previously observed in Progeria transcriptomics analysis (*Ikegami et al., 2020*) However, after batch effect correction (materials and methods *Zhang et al., 2020a*), the first principal component was able to segregate the samples depending on progeria/non progeria origin (*Figure 1—figure supplement 1B*) which enabled direct comparisons between patients and controls from different age groups which originated from different sources.

## Gene ontology analysis of transcriptional changes reveals eight clusters of biological activity affected in HGPS

To derive sets of genes differentially regulated in Progeria, we divided the patient samples into Young (0–8 years old) and Teen (>13 years old) categories based on the idea that different developmental processes take place in these age groups. We then compared each patient group to age-matched controls, middle-aged adults, and older adults and selected up and down-regulated genes (FDR adjusted *P*-value <0.001) for gene ontology analysis. Teenage HGPS patients showed the smallest number of genes changing expression levels when compared to age-matched controls, but the highest number of genes up/down regulated when compared to healthy middle-aged or old adults (*Table 1* and *Table 1—source data 1*). Importantly, comparisons between the normal children cohort and the adult normal controls yield few significant changes, suggesting that HGPS is the driver of the differences observed among cohorts.

Overall, relevant gene ontology terms identified in the study pertain to eight functional clusters: DNA maintenance and Epigenetics (*Figure 1*), Repair and Extracellular matrix (*Figure 2*), Bone (*Figure 3*), Adipose Tissue (*Figure 4*), Blood Vessels (*Figure 5*), and Muscle (*Figure 6*).

The largest number of differentially expressed genes (Nine hundred and seventy) were related to the biological processes of epigenetic programming and DNA maintenance (*Figure 1A* and *Figure 1—source data 1*). Gene ontology pathways included in this cohort include epigenetic regulation of gene silencing, senescence-associated heterochromatin, DNA repair, DNA recombination, and histone methylation/modifications. This is in line with extensive previous evidence of epigenetic and DNA repair misregulation in Progeria (*Aguado et al., 2019*; *Gonzalo and Coll-Bonfill, 2019*; *Misteli and Scaffidi, 2005*). These genes were predominantly over-expressed in young progeria patients when compared to all controls (*Figure 1B,C*) and between teenaged patients compared to their age matched control and adults (*Figure 1B,D*).

The next biological process with the most misregulated transcription among age groups was tissue repair. It bears mentioning that most of the nucleosome-associated proteins and histones identified in the DNA maintenance cluster, as previously described, overlap with the repair GO terms. In all, 585 genes are differentially expressed between cohorts (*Figure 2A*), with HGPS patients predominantly overexpressing targets pertaining to the organization of the extracellular matrix (145 genes,

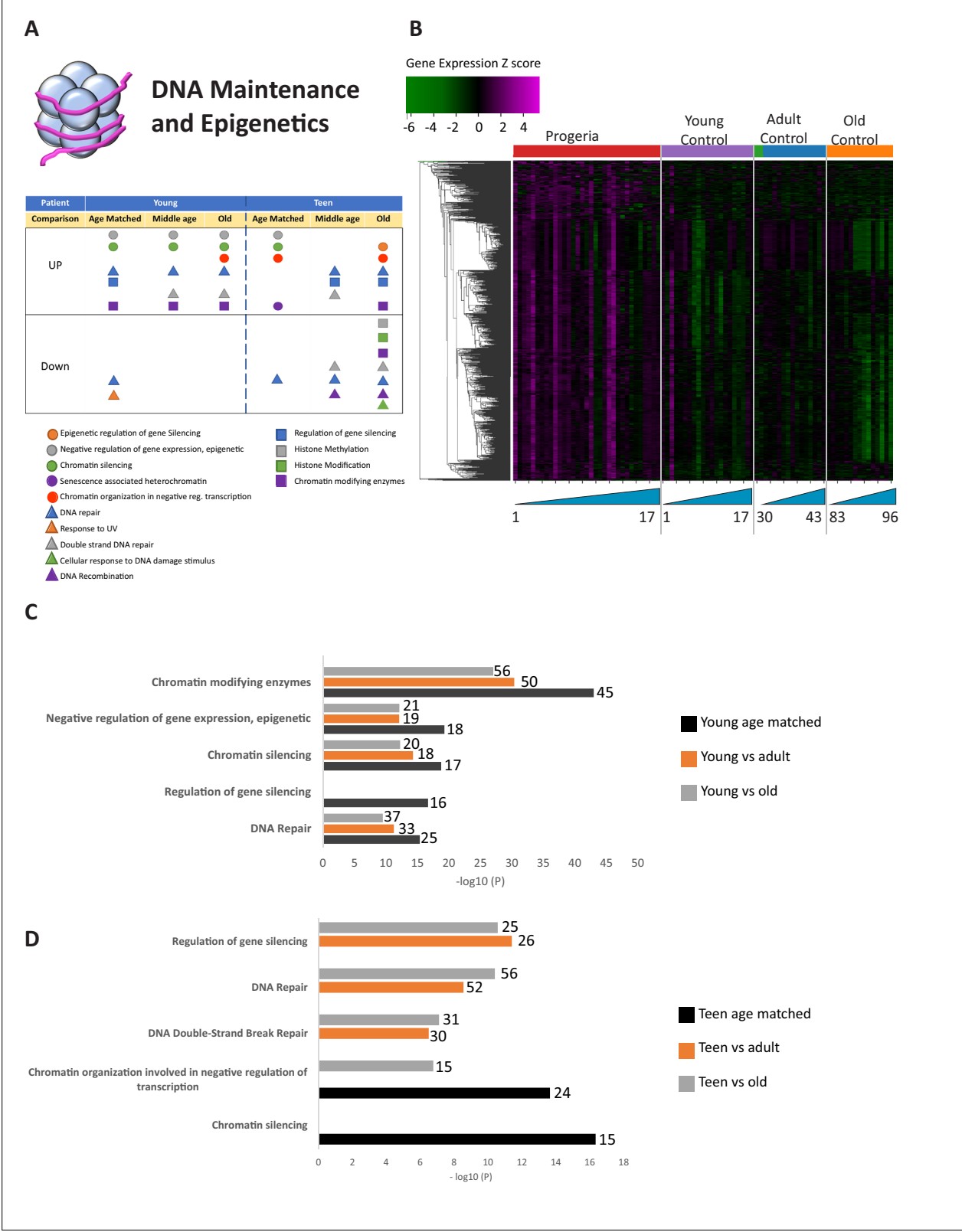

**Figure 1.** Transcriptional misregulation in the DNA Maintainance and Epigenetics functional categories. (**A**) Summary table of processes related to DNA maintenance and epigenetics, represented as transcriptional up or downregulation based on RNA-seq of young/teenager progeria patient derived fibroblasts compared to age matched, middle age or old control patients. (**B**) Heat map of RNA-seq transcriptome analysis for 976 selected genes related to DNA maintenance and epigenetics. The heat map shows per-gene z-score computed from batch effect corrected log2 read count

*Figure 1 continued on next page*

*Figure 1 continued*

values, genes in rows and 29 patient samples (progeria and young/adult/old control) organized in columns. Genes were hierarchically clustered based on Euclidean distance and average linkage. Within each cohort, columns are organized by patient age. (**C**) Comparison of young progeria patients versus middle age or old donor control fibroblasts. Enriched ontology clusters for upregulated genes related DNA maintenance and epigenetics, as characterized by Metascape analysis. Metascape reports p-values calculated based on the hypergeometric distribution. (**D**) Comparison of teen-aged progeria patient fibroblast versus old donor control fibroblasts. Enriched ontology clusters for up regulated genes related to DNA maintenance and epigenetics, as characterized by Metascape analysis.

The online version of this article includes the following source data and figure supplement(s) for figure 1:

**Source data 1.** RNA-seq results: normalized and batch corrected sequencing counts for all samples in this study.

**Figure supplement 1.** Batch clustering correction.

*Figure 2B*). Interestingly, signatures related to coagulation, Bone Morphogenic Protein signaling, and response to wound healing are all downregulated in young HGPS patients when compared to adults (*Figure 2C*).

In this analysis of fibroblasts, we also identified misregulation of gene targets typically associated with the biology of mesenchymal tissue. Specifically, we see misregulation of transcription of genes involved in bone (165 genes. *Figure 3*), fat (261 genes. *Figure 4*), blood vessel homeostasis (131 genes. *Figure 5*), and muscle (261 genes. *Figure 6*). All these lineages have been observed as compromised in HGPS patients, showing phenotypes like osteopenia, early atherosclerosis, and lack of subcutaneous fat deposition. (*Gordon et al., 2011*; *Hamczyk et al., 2018*; *Xiong et al., 2013*).

Bone biology mis-regulation appears to be stratified into two distinct hubs: downregulation of pathways related to calcium homeostasis and transport and upregulation of pathways involved in cell differentiation of chondrocytes and osteoblasts (*Figure 3A,B* and *Figure 3—source data 1*). Comparing young patients' transcriptional profile to that of old adult normal controls shows upregulation of genes involved in late osteogenic commitment. Such is the case of ontology terms that include osteoblast differentiation, ossification, and chondrocyte differentiation (*Figure 3C*). In parallel, genes included in pathways related to cation homeostasis, particularly calcium, are downregulated (*Figure 3D*). This comparison is similar to that of young patients and normal middle-aged adult controls (*Figure 3E*).

Lipid homeostasis and transport pathways are downregulated in all comparisons between samples from young patients. Young HGPS samples show downregulation in pathways involving fat cell differentiation when compared to age-matched controls, in line with observed fat deposition defects in patients (*Revêchon et al., 2017*; *Figure 4A–C*).

Gene ontology terms related to blood vessels show upregulation of blood vessel morphogenesis, but a decrease of blood vessel maturation when comparing samples from young patients to either age-matched or old controls (*Figure 5A–C*). This phenotype persists in comparisons between teen aged patients and their age-matched controls (*Figure 5D*). Further, in a possible contribution to the circulatory defects observed in HGPS patients, both young and teen-aged patient samples show a marked upregulation of genes in pathways related to muscle development and cardiac muscle differentiation and development when comparing both HGPS cohorts against adult controls (*Figure 6A–C*).

## Age stratification of young patients highlights differences in HGPS gene misregulation across childhood

To further refine our findings related to young age patients and the mesenchymal phenotypes observed, we stratified the young HGPS patients into two age groups: early infancy (0–3 years old) and children (4–7 years old), comparing these cohorts to age matched, middle age and old controls as before. The total numbers of genes up and down regulated between comparisons are described in *Table 2* and *Table 2—source data 1*. In this analysis, about forty percent of genes upregulated in early infant patients when compared against their age matched controls are related to epigenetic modifications as described in the previous analysis (*Figure 1*). Specifically, the GO term 'HDACs deacetylate histones' shows the highest enrichment in this group. These same genes are represented in the comparison with the middle age-old adult control groups.

In contrast, the comparison of patients aged 4–7 years to their age matched controls show an enrichment for ossification and calcium homeostasis. Out of the cohort of genes identified as

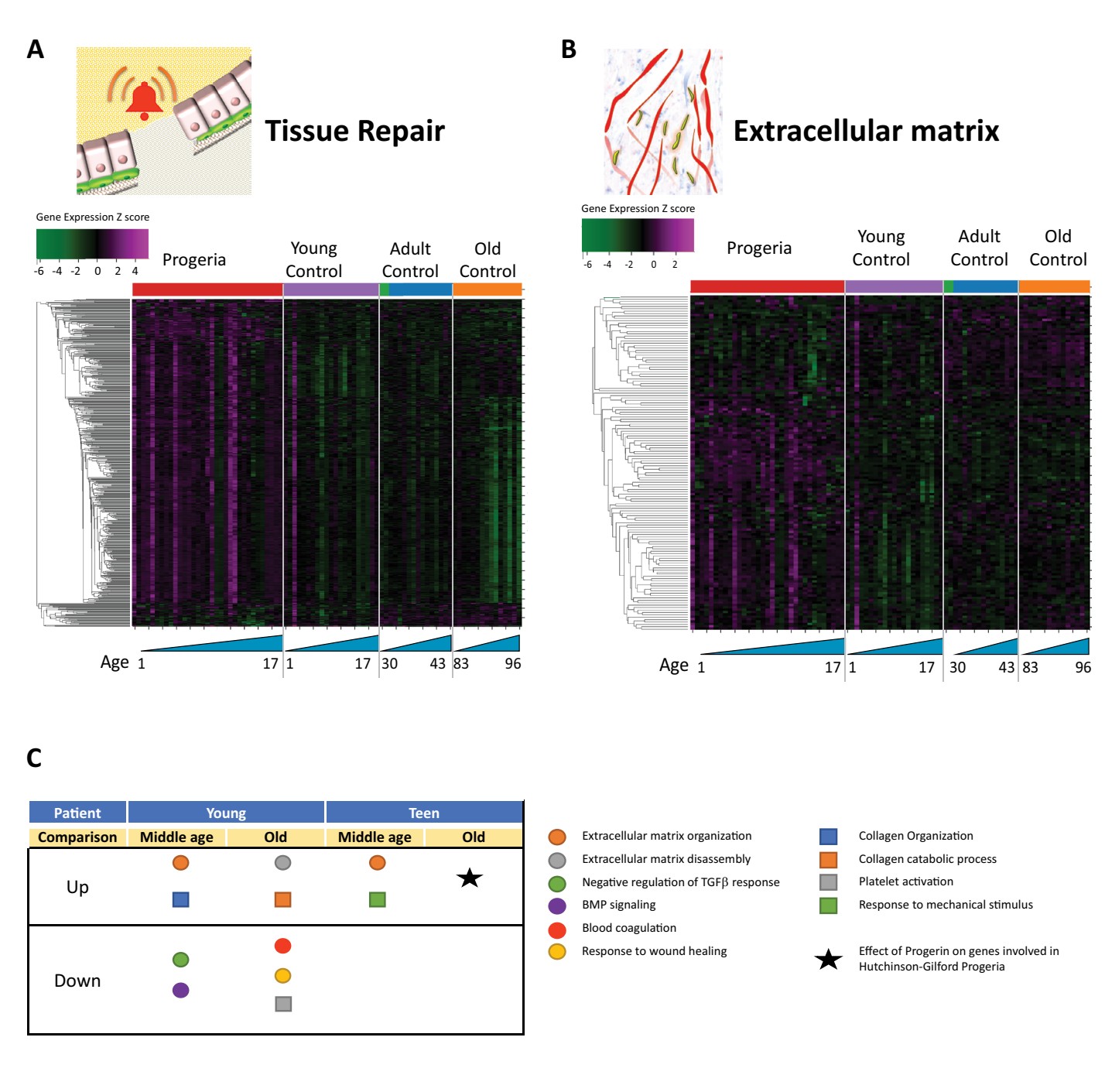

**Figure 2.** Transcriptional misregulation in tissue repair and extracellular matrix functional categories. (**A**) Heat map of RNA-seq transcriptome analysis for 585 selected genes related to repair. Data presented as in *Figure 1B*. (**B**) Heat map of RNA-seq transcriptome analysis for 145 selected genes related to extra cellular matrix. Data presented as in *Figure 1B*. (**C**) Summary table of processes related to repair and extra cellular matrix organization, represented as up or downregulation in transcription based on RNA-seq of young/teenager progeria patient derived fibroblasts, compared to middle age or old control patients. Age comparisons that yielded no significant results in relevant categories are not shown.

downregulated in this comparison (102), about 10% are related to early chondrogenesis events. Among those, parathyroid hormone related protein (PTHLH), insulin like growth factor (IGF1), bone morphogenic protein receptor 1B (BMPR1B), and collagen 10a1 (COL10a1), are an integral part of early commitment and control necessary for the triggering of endochondral ossification (*Bradley and*

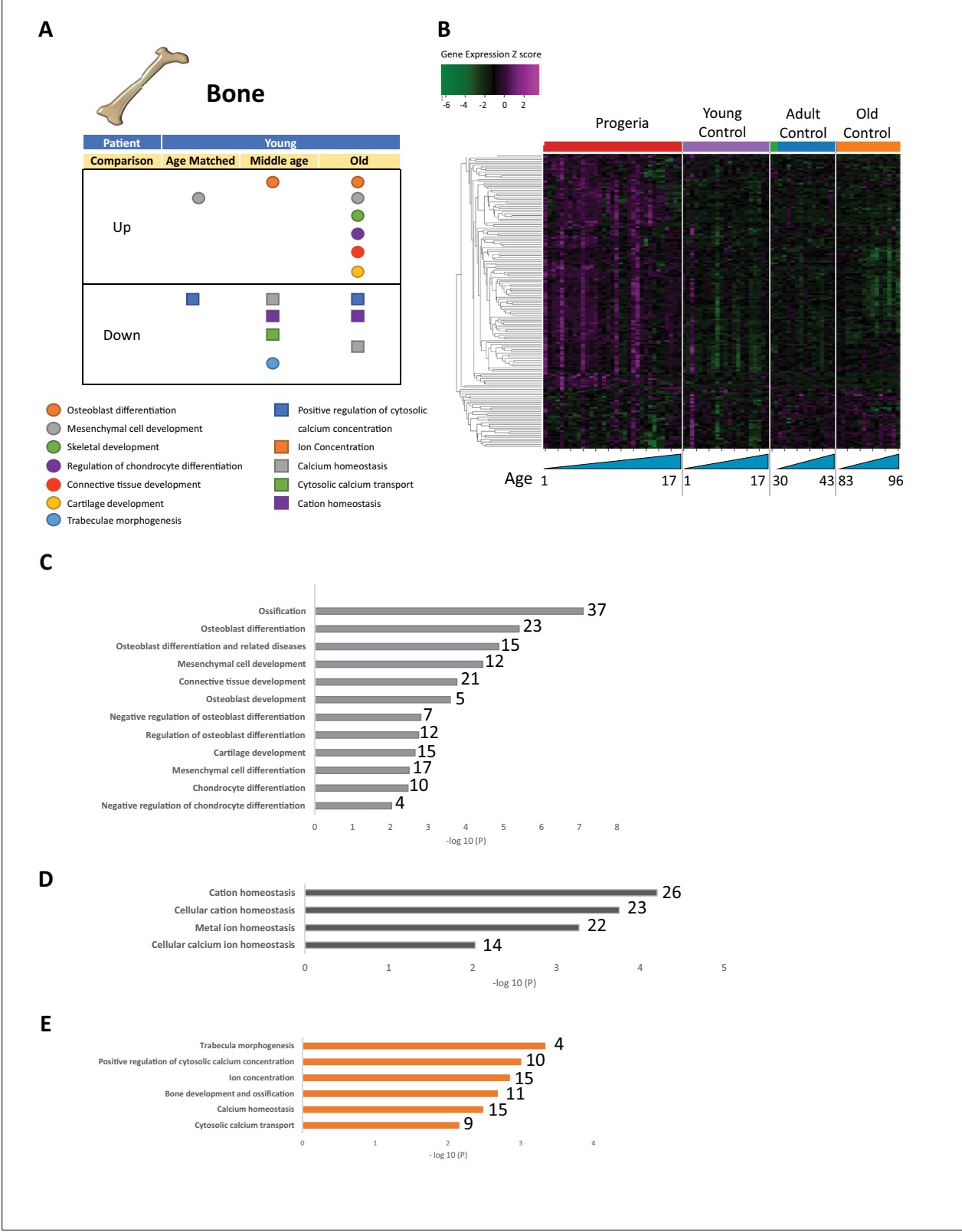

**Figure 3.** Transcriptional misregulation for genes related to bone development. (**A**) Summary table of processes related to bone and cartilage development and homeostasis represented as up or downregulated transcription based on RNA-seq of young progeria patient derived fibroblasts compared to age matched, middle age or old control patients. (**B**) Heat map of RNA-seq transcriptome analysis for 165 selected genes related to bone and cartilage development. Data presented as in *Figure 1B*. (**C**) Comparison of young progeria patients versus old donor control fibroblasts.

*Figure 3 continued on next page*

*Figure 3 continued*

Enriched ontology clusters for upregulated genes related bone and cartilage development and homeostasis, as characterized by Metascape analysis. (**D**) Comparison of young progeria patients versus old donor control fibroblasts. Enriched ontology clusters for downregulated genes related to cation homeostasis, as characterized by Metascape analysis. (**E**) Comparison of young progeria patients versus middle age control fibroblasts. Enriched ontology clusters for downregulated genes related to bone development and homeostasis, as characterized by Metascape analysis.

The online version of this article includes the following source data for figure 3:

**Source data 1.** Gene expression analysis (Z-scores) for 164 genes related to bone development.

*Drissi, 2010*; *Green et al., 2015*; *Maruyama et al., 2010*). Genes related to ossification and calcium homeostasis were also identified as downregulated in this cohort. Further, upregulated genes in age-matched and adult comparisons include BMP4, and several genes related to WNT5a biology which play an important role in skeletal development (*Figure 7A–C*).

## Genes of interest are impacted by chromatin compartment switches or abnormal lamina associated domains (LADs)

Chromosome conformation capture was performed on skin fibroblasts from parents of HGPS patients (Mother AG03257, Father HGADFN168), from 8-year-old male and female HGPS patients (HGADFN167 and AG11513), and from an age matched (8-year-old male) healthy control (GM08398) (*Appendix 1— table 4*). Genome-wide contact maps for all of these cells show globally similar patterns of chromosome conformation, with the exception of a translocation between chr3 and chr11 in AG11513 cells (*Figure 8A*). At a whole chromosome folding level, we note that progeria fibroblasts show a decrease in telomere interactions and an apparent loss of 'Rabl' like structure (*Figure 8B*). This type of Rabl structure loss was previously observed after DNA damage in fibroblasts (*Sanders et al., 2020*). We find that topologically associating domain (TAD) structure is preserved in Progeria fibroblasts, with even an increase in TAD boundary strength as compared to healthy parent controls (*Figure 8C*). A similar increase in TAD boundary strength was previously found in senescent and progerin-expressing human mesenchymal progenitor cells (*Liu et al., 2022*). Increased TAD boundary strength was also previously observed in DNA damaged fibroblasts (*Sanders et al., 2020*), suggesting that some of these features of genome structure may relate to the increased constitutive levels of DNA damage observed in these patient fibroblasts.

Using principal component analysis of Hi-C data at 250 kb-resolution, we classified genomic regions into open euchromatin (A) or closed heterochromatin (B) spatial compartments according to positive and negative values of the first eigenvector, respectively. Since accelerated senescence has been previously observed in progeria fibroblasts (*Bridger and Kill, 2004*; *Wheaton et al., 2017*), we performed compartment analysis on cells belonging to a father-child pair, at early and late passages. Samples were collected for the HGADFN168 cells (Father) at passages 12 and 27, and for the HGADFN167 (Child) at passages 12 and 19. Our results show that in the parent fibroblasts there are small changes in compartment strength (the degree of preference for interactions within the same compartment vs. between different compartments). In contrast, there is a predominant loss of A compartment strength in the late passage HGPS cells (*Figure 8E*). Interestingly, although the compartmentalization strength appears to be altered, the genome-wide compartment identity remains consistent among cohorts, with small changes in compartment identity appearing sporadically (*Figure 8D*). The loss of compartment strength at later passages in Progeria cells is consistent with previous observations, though this more deeply sequenced, and less noisy dataset shows that compartments are more preserved in Progeria cells than previously observed (*McCord et al., 2013*).

To test whether the up or downregulation of genes identified through this study relates to compartment switching, we evaluated the compartment identity strength of all up- or down-regulated genes, based on their genomic coordinates, in the healthy and patient datasets. We found that genes that are upregulated in HGPS tend to be located in the A compartment in both HGPS and WT fibroblast samples, suggesting that many expression increases do not require chromosome compartment shifts (*Figure 8F–H*). In contrast, a subset of the genes that are downregulated in the 0–7 HGPS fibroblasts vs. normal age- matched or adult fibroblasts also shift toward the B compartment in HGPS samples compared to healthy adults. (*Figure 8F–H*).

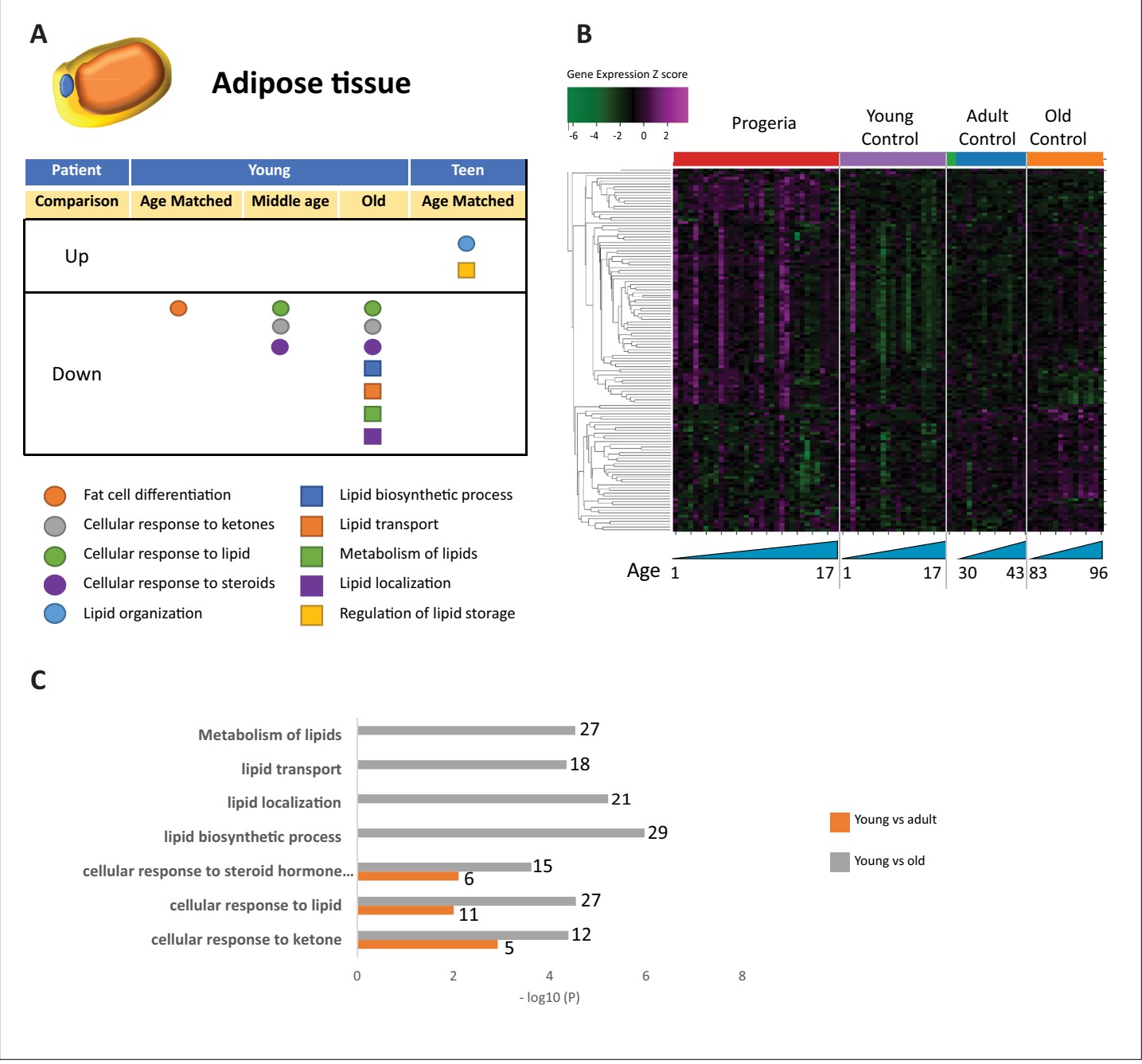

**Figure 4.** Transcriptional missregulation in genes related to adipose tissue function and development. (**A**) Summary table of processes related to fat cell differentiation and lipid metabolism, represented as up or downregulation in transcription based on RNA-seq of young/teenager progeria patient derived fibroblasts, compared to age matched, middle age, or old control patients. (**B**) Heat map of RNA-seq transcriptome analysis for 134 selected genes related to fat cell differentiation and lipid metabolism. Data presented as in *Figure 1B*. (**C**) Comparison of young progeria patients versus middle age or old donor control fibroblasts. Enriched ontology clusters for upregulated genes related to fat and lipid metabolism, as characterized by Metascape analysis.

To test whether the up or downregulation of genes identified through this study relates to abnormal distribution of lamin associated domains, we compiled the genomic coordinates of these genes with published Lamin ChIP-seq and DamID-seq data (*Dekker et al., 2017*; *McCord et al., 2013*). We observe that differences between all groups are significant, with HGPS Lamin A association values markedly higher than their control counterparts (*Figure 8—figure supplement 1A and B*, Kruskal-Wallis p<0.001). However, we found that this shift in Lamin association is true throughout all genomic

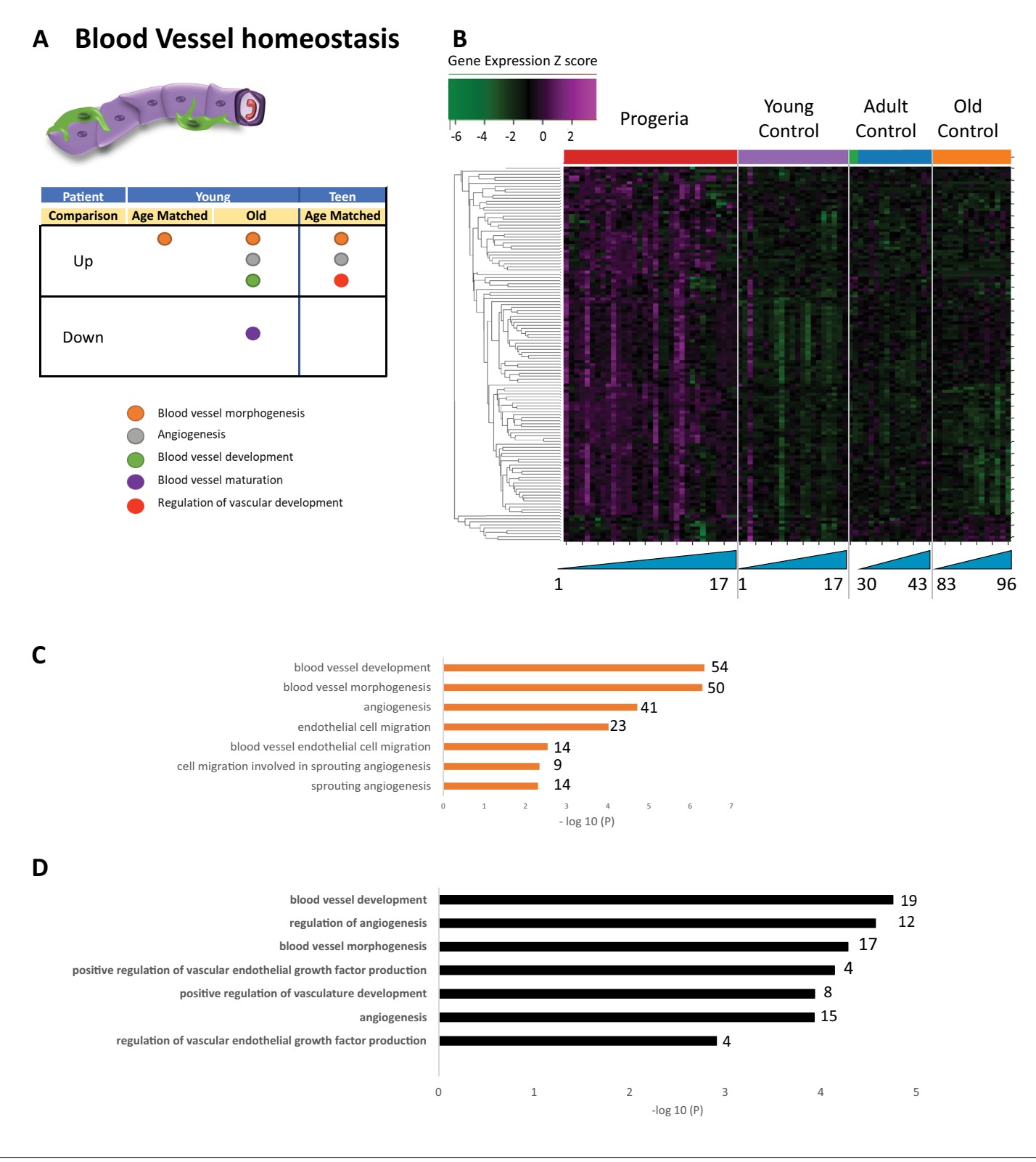

**Figure 5.** Transcriptional missregulation in genes related to blood vessel homeostasis. (**A**) Summary table of processes related to blood vessel homeostasis, represented as up or downregulation in transcription based on RNA-seq of young/teenager progeria patient derived fibroblasts, compared to age matched, middle age, or old control patients. (**B**) Heat map of RNA-seq transcriptome analysis for 131 selected genes related to blood vessel homeostasis. Data presented as in *Figure 1B*. (**C**) Comparison of young progeria patients versus middle-aged donor control fibroblasts.

*Figure 5 continued on next page*

*Figure 5 continued*

Enriched ontology clusters for upregulated genes related to blood vessel development, as characterized by Metascape analysis. (**D**) Comparison of teen-aged progeria patient fibroblast versus age-matched control fibroblasts. Enriched ontology clusters for upregulated genes related to blood vessel development, as characterized by Metascape analysis.

regions, with an overall dampening of the lamin association signal in HGPS patients (*Figure 8—figure supplement 1C*) and small regions switching from associated to dissociated from the nuclear lamina. Comparing normal fibroblast cells HGADFN168 (belonging to the father of an HGPS patient) and HFFc6 (human foreskin fibroblast), we observe that both show a characteristic bimodal distribution in LAD intensity around markedly positive (LAD) and negative (non-LAD) values (log2 Lamin/input). In contrast, LAD values for the HGPS fibroblasts HGADFN167 show a normal distribution around zero: the intensity of both types of association is reduced in HGPS cells, consistent with an overall misregulated interaction between chromosomes and the nuclear lamina (*Figure 8—figure supplement 1D*).

Certain key genes of interest related to osteogenesis and chondrogenic proliferation and identified as differentially regulated by transcriptomics show notable concordant alterations in compartment identity or lamina association. For example, bone morphogenic protein 4 (BMP4), which is upregulated in Progeria (*Figure 7C*), also shows a marked shift toward the A compartment in patient fibroblasts compared to both adult and age matched WT fibroblasts (*Figure 9*). In turn, Frizzled-4 (FZD4), which is upregulated in HGPS and involved in skeletal development, does not shift compartments but shows an erosion of Lamin association (*Figure 9—figure supplement 1*). Similarly, the gene that encodes parathyroid hormone-like hormone (PTHLH), which is downregulated in HGPS, is in a conserved A compartment in both healthy and HGPS cells but shows an increase in Lamin association in patient cells (*Figure 9—figure supplement 1*). Consistent with the observation that compartment shifts were more likely to be found alongside gene downregulation, we also identified a set of key genes in the mesenchymal lineage that were both downregulated and shifted toward the B compartment in HGPS (*Figure 9—figure supplement 2*).

## Discussion

In this study, we present a comprehensive analysis of the transcriptome of HPGS patient derived fibroblasts, stratified by age, compared to their age-matched controls. Since progeria patients present with accelerated aging, we also compared this data to that derived from middle-aged and old adults. Strikingly, HPGS expression profiles do not phenocopy gene ontology analysis from either set of adult controls, suggesting an altered aging-like paradigm.

Given the critical role that Lamin A plays in nuclear structure, formation of heterochromatin, and subsequent gene silencing (*Lammerding et al., 2004*; *Leemans et al., 2019*), it is not surprising that one of the main biological processes affected in our study, as per gene ontology analysis, is DNA maintenance and epigenetics. Upregulation of genes in the HDAC family is the largest discriminator between Progeria and age-matched normal controls in the 0- to 3-year-old cohort and 4- to 7-year-old age groups, and this upregulation persists in comparisons with the adult controls. Our data suggests an upregulation of genes belonging to ontology terms such as negative epigenetic regulation of genes, chromatin silencing, DNA repair, double strand DNA repair and DNA recombination across all age comparison with young patients. Overexpression of these genes are a potential overcompensation mechanism for defects found in DNA repair (*Aguado et al., 2019*; *Gonzalo and Kreienkamp, 2015*; *Komari et al., 2020*; Reviewed by *Misteli and Scaffidi, 2005*). In older HGPS samples, downregulation of genes related to histone methylation, histone modification, chromatin modifying enzymes, and DNA recombination was also observed.

Concomitant with changes in gene expression of epigenetic factors, Progeria fibroblasts show alterations in chromosome structure, as has been previously observed (*McCord et al., 2013*). The increased resolution and clarity of Hi-C data presented here enables us to describe these alterations more clearly. Spatial compartmentalization is weakened in Progeria patients with increasing passages consistent with microscopically observed loss of heterochromatin (*Goldman et al., 2004*). Unlike compartments, topologically associating domain (TAD) boundaries are preserved during in Progeria cells, even at higher passages. When we compare gene expression changes to chromosome structure changes, we observe that many genes consistently upregulated across Progeria patients are in the

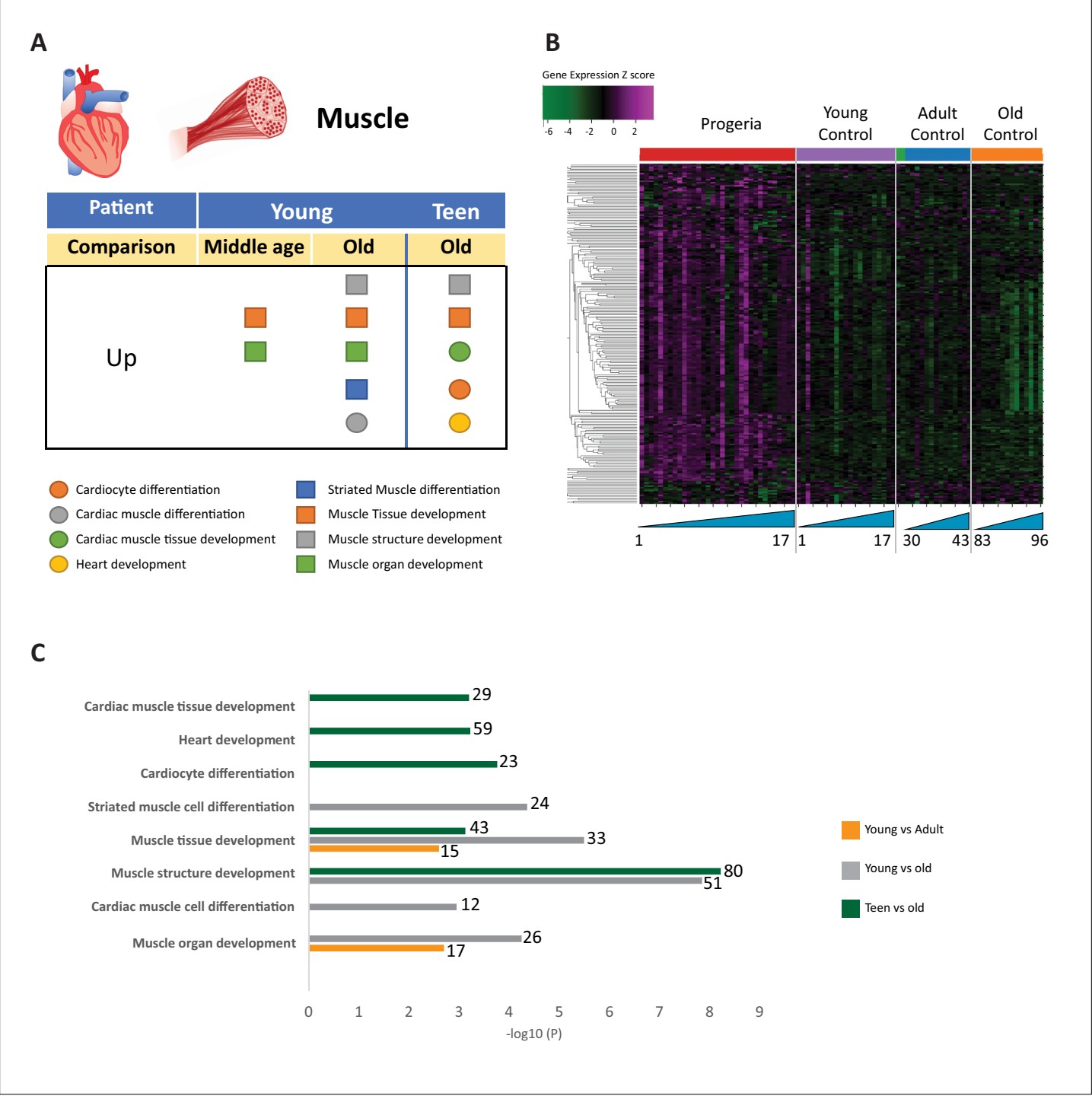

**Figure 6.** Transcriptional missregulation in genes related to muscle function. (**A**) Summary table of processes related to muscle and cardiac muscle development, represented as up or downregulation in transcription based on RNA-seq of young/teenager progeria patient derived fibroblasts, compared to middle age or old control patients. (**B**) Heat map of RNA-seq transcriptome analysis for 261 selected genes related to muscle development. Data presented as in *Figure 1B*. (**C**) Comparison of young progeria patients versus middle-aged and old donor control fibroblasts and of teen-aged progeria patients versus old donor controls. Enriched ontology clusters for upregulated genes related to muscle development, as characterized by Metascape analysis.

**Table 2.** Number of up/down-regulated genes in young patient cohort comparisons.

| Comparisons | Upregulated genes | Downregulated genes |
|---|---|---|
| Early Infant – Age matched | 113 | 25 |
| Early Infant – Middle aged adult | 540 | 172 |
| Early Infant – Old adult | 646 | 388 |
| Children - Age matched | 203 | 102 |
| Children - Middle aged adult | 173 | 184 |
| Children - Old adult | 305 | 394 |

The online version of this article includes the following source data for table 2:

**Source data 1.** Metascape outputs for gene ontology analysis.

open, A compartment across all Hi-C samples, suggesting that these gene expression alterations are not associated with dramatic chromosome structural change. Downregulated genes are more likely to exhibit switching into the B compartment in Progeria patients. We find that certain genes relevant to the mesenchymal lineage show gene expression changes that are concordant with compartment switches and lamin association changes. It is possible that these changes are more directly influenced by mutant lamin and lead to downstream effects on genes without observed chromosome structure changes.

## A compromised transcriptional landscape, related to developmental milestones, points towards a compromised mesenchymal stem cell niche

We further describe up and down regulated biological function clusters that could play a detrimental role in bone, fat, joint and vascular homeostasis. It has been proposed that progerin accumulation in the nucleus results in a defect in the mesenchymal stem cell lineage which gives rise to osteoblasts, chondrocytes, adipocytes, pericytes, and myocytes (Reviewed by *Andrzejewska et al., 2019*). Overall, our results are in concordance with previous reports of transcriptional misregulation in mesenchymal lineages (*Csoka et al., 2004*), but the age group comparisons we present here further refine these observations to a temporal effect on lineage commitment.

iPSC models in which HGPS fibroblasts were reprogrammed to stem cells, and further differentiated into mesenchymal stem cells lineages were characterized by nuclear dysmorphia and increased DNA damage, but resulted in confounding results on differentiation, showing either limited differentiation potential, or no significant changes (*Crasto and Di Pasquale, 2018*; *Xiong et al., 2013*; *Zhang et al., 2011*). Interestingly, the most common finding in HGPS-derived iPSCs further differentiated into other lineages is premature senescence and the presence of progerin and misshapen nuclei (Reviewed by *Lo Cicero and Nissan, 2015*). In a more direct approach, overexpression of progerin in umbilical cord derived MSCs and other MSC systems resulted in a reduced capacity of differentiation into chondrogenic, osteogenic and adipogenic capacity in vitro (*Mateos et al., 2013*) and deficient proliferation and migration (*Pacheco et al., 2014*). In the context of these collective findings, our age-stratified gene expression comparisons in fibroblasts may shed light on whether this abnormal cell fate regulation occurs predominantly in a particular lineage.

## Failed arrest of chondrocyte hypertrophy as a potential mechanism of MSC depletion in HGPS

Transcriptional upregulation of several members of the WNT5a signaling cascade and downregulation of expression of PTHLH, in early age comparisons described in this study, point to an essential defect in early endochondral ossification control and osteogenesis. Specifically, WNT signaling has been previously implicated as a connection between progeria-like syndromes and a defective deposition of extra cellular matrix, essential in bone development (*Andrade et al., 2017*; *Green et al., 2015*; *Hernandez et al., 2010*). Not unlike the phenotypes observed in HGPS patients, bone defects are also present in murine models for HGPS, with young mice (4-week-old) showing decreased trabecular thickness and number, bone volume and decreased mineral density (*Hernandez et al., 2010*). Not

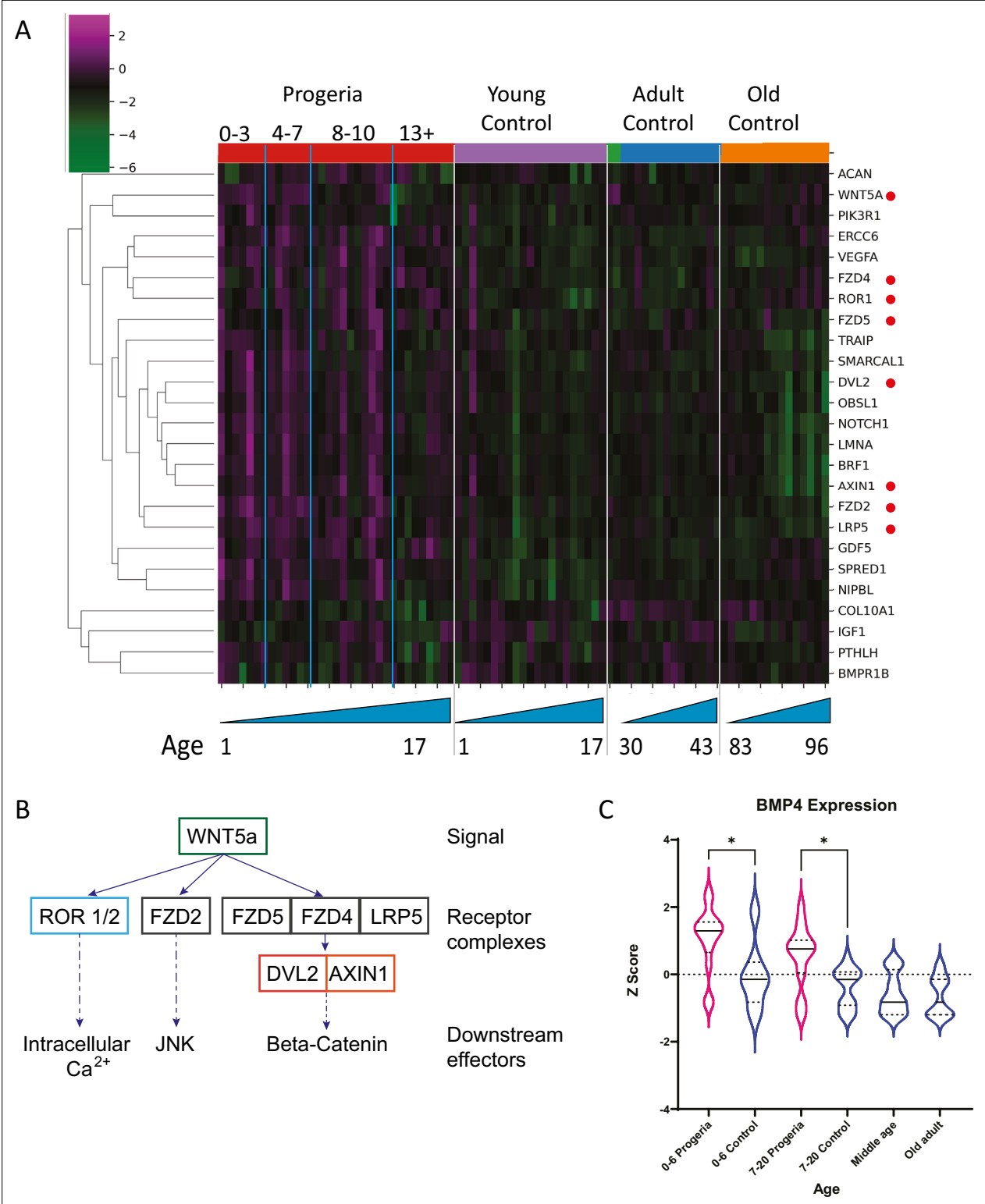

**Figure 7.** Transcriptional missregulation in genes related to endochondral ossification. (**A**) Heat map of RNA-seq transcriptome analysis for 25 selected genes related to endochondral ossification. The heat map shows per-gene z-score computed from batch effect corrected log2 read count values, genes in rows and 29 patient samples (progeria and young/adult/old control) organized in columns. Genes were hierarchically clustered based on Euclidean distance and average linkage. Blue lines separate young infants (0–3 y/o) from children (4–7 y/o) and older children (8–10 y/o). Genes related to WNT5a biology highlighted (red circle). (**B**) Abridged signaling pathway for WNT5a, highlighting the roles of the genes whose transcription is affected in young HGPS patients compared to their aged-matched controls. (**C**) BMP4 expression, also identified as significantly upregulated in the analysis of all HGPS

*Figure 7 continued on next page*

*Figure 7 continued*

patients (*Figure 3*) differs between Progeria (pink) and control (blue) samples. (* indicates p<0.05 by Kruskal Wallis test; dotted lines within the violin plots indicate 25th, median, and 75th percentiles).

surprisingly, murine pre-osteoblasts forced to express mouse progerin failed to mineralize and show decreased levels of Runx2 expression and alkaline phosphatase upon osteoblastic induction (*Tsukune et al., 2019*). Further, Lamin A knockdown inhibits osteoblast proliferation and impairs osteoblast differentiation in an MSC in vitro differentiation model (*Rauner et al., 2009*).

Unlike other MSC fates, like adipogenesis, which occurs downstream of a positive energy intake all through life, bone formation occurs in a carefully orchestrated manner, at specific timepoints during development and early life. Secondary endochondral ossification that occurs in long bones like the humerus and femur, initiate in early childhood, activate intermittently, and finalize with the fusion of the growth plate during the teenage years (*Diméglio et al., 2005*; *Xie and Chagin, 2021*; *Zoetis et al., 2003*). This process is tightly regulated by several signaling networks aimed at balancing longitudinal growth of the bone with maintenance of precursor cells, like MSCs and quiescent chondrocytes. The transcriptional repression of PTHLH and upregulation of WNT5a in this niche would result in early depletion of the latter, as cells continue down a pathway of chondrocyte proliferation and hypertrophy unimpeded (*Bradley and Drissi, 2010*; *Olsen et al., 2000*; *Usami et al., 2016*). The deficit in clavicular development in HPGS patients could provide some insight as to the spatiotemporal consequences of MSC/chondrocyte pool depletion. The clavicle is unique because while it is the first bone to start ossification during the embryonic period (*Ogata and Uhthoff, 1990*), it is the last to complete the process. In early events, around the fifth week of gestation, the two primary ossification centers, formed from mesenchymal tissues, fuse to form the middle of the clavicle. The complete ossification of the clavicular epiphysis occurs via endochondral ossification during the teenage years, with medial ossification beginning at the onset of puberty (*Ferguson and Scott, 2016*; *Langley, 2016*). In turn, bone deposition at the terminal ossification center, must occur in adolescents between 11 and 16 years of age, for the process to successfully proceed to epiphyseal plate ossification in early adulthood. (*Schulz et al., 2008*). Taken together, our observations point towards a 'fail to arrest' phenotype that results in over-commitment towards the hypertrophic chondrocyte lineage in very young HGPS patients. We hypothesize that this results in a premature depletion of the chondro-osteoprogenitor pool and, at later stages, of the bone marrow-derived mesenchymal stem cells that feed into this niche later in life result in the incomplete endochondral ossification of the claviculae of HGPS patients (*Video 1*).

## Deficient repair responses exacerbate the depletion of tissue resident MSC pools

A secondary function of tissue resident MSCs is to aid in repair during wounding and disease.

Our data shows that dermal fibroblasts derived from HPGS patients present with abnormal repair, including downregulated signatures of wound healing, hemostasis, and BMP signaling. These observations recapitulate previous reports of HGPS patient-derived skin precursors, which can differentiate intro fibroblasts and smooth muscle cells (smooth muscle alpha actin positive – myofibroblasts). In the precursor stage, these cells express low levels of progerin in vivo and in vitro, but differentiation into the lineages increases progerin expression and deposition in the nucleus (*Wenzel et al., 2012*). In concordance with our findings, a progeria murine model, deficient in the downstream processing of lamin A, shows abnormal skin wound repair, with prolonged time to wound closure, poor vasculogenic signaling and angiogenesis, which includes a limited mobilization of bone-marrow derived progenitor cells (*Butala et al., 2012*). Related to this phenotype, the LMNAΔ9 murine progeria model for HPGS show a significant decrease in postnatal fibroblast proliferation, with accelerated senescence levels in cell lines derived from kidney, lung, skin, and skeletal muscle (*Hernandez et al., 2010*). Further, a decreased epidermal population of adult stem cells was observed in another murine model of HGPS (*Rosengardten et al., 2011*), which are also deficient for skin wound repair.

Overall, growth, development, and maintaining homeostasis place a considerable burden on the mesenchymal stem cell pool and its many lineages. Under a paradigm of biology of priority during development the organism's structural integrity in the shape of bones and joints, the homeostasis of blood microvessels – regulated by pericytes – should take precedence to fat deposition (*Figure 10A*).

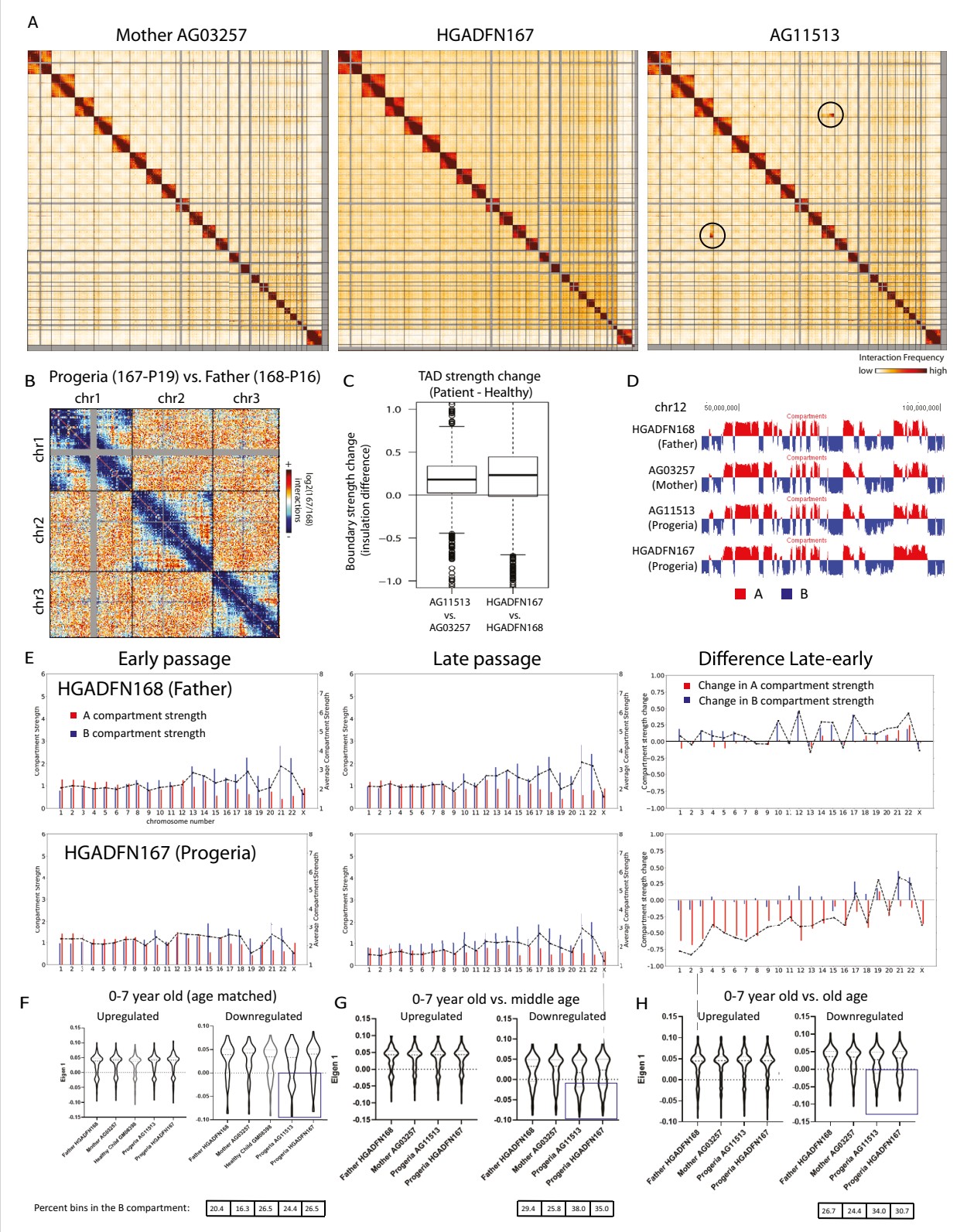

**Figure 8.** Changes in chromatin architecture in HGPS cells: translocations, compartment strength and identity, and correlation to genes of interest. (**A**) 2.5 Mb Hi-C heatmaps for AG03257-P7 (Mother, WT), and HGPS patients (HGADFN167-P19 and AG11513-P7) Translocations between chromosomes appear as high interaction frequency regions (red) away from the diagonal. A translocation between chromosomes 3 and 11 is apparent in AG11513 cells (circle). (**B**) Log ratio of 2.5 Mb contact frequency in Progeria (167-P19) vs. healthy father (168-P16). Loss of telomere interactions is notable as blue

*Figure 8 continued on next page*

*Figure 8 continued*

patches in the corner of each chromosome. (**C**) TAD boundary strength boxplots calculated using the InsulationScore approach between early (left) and late (right) passage Progeria cells minus their respective controls. Boxes represent the upper and lower quartiles with the center line as the median. Upper whiskers extend 1.5×IQR beyond the upper quartile, and lower whiskers extend either 1.5×IQR below the lower quartile or to the end of the dataset. (**D**) Plots of the first eigenvector for a section of chromosome 12, obtained from principal component analysis (PC1) of 250 kb binned Hi-C data for control fibroblasts (Mother AG03257, Father HGADFN168) and HGPS fibroblasts (HGADFN167 and AG11513). Compartment identity remains predominantly unchanged (A compartment: Red, B compartment: Blue). (**E**) Graphs showing the A-A compartment interaction strength (red) and B-B compartment interaction strength (blue) within each chromosome for related father and child cell lines (HGADFN168, HGADFN167). Samples were collected a both early (left; P12) and late passages (middle; P19 for Progeria and P27 for father). Comparison between the two samples (right) shows that the HPGS cell line shows a marked decrease in A-A compartment interaction strength in late passages in the majority of chromosomes. (**F**) Eigen 1 values represent the compartment identity (same as plotted in D) for genes identified in this study as upregulated (left) or downregulated (right) in the 0–7 year-old age-matched comparison. While differences between groups are not significant overall (Kruskal-Wallis), a subset of downregulated genes appear to be changing conformation to a B compartment in progeria samples (box). Violin plots for the global distribution of values; median denoted by a thick dashed line, 25th and 75th percentiles highlighted as thin dashed lines. Percentages of genes in the B compartment are indicated in the box below downregulated gene graph. (**G**) Compartment identity for genes identified in this study as upregulated (left) or downregulated (right) in the 0–7 year-old HGPS samples compared to normal middle-aged controls. (**H**) Compartment identity for genes identified in this study as upregulated (left) or downregulated (right) in the 0–7 year-old HGPS samples compared to old-aged controls.

The online version of this article includes the following figure supplement(s) for figure 8:

**Figure supplement 1.** Genes of interest vs LADS.

We propose that these defects in repair in Progeria patients strain the mesenchymal stem cell pools from which repair reactive stroma originates, which have been characterized as both pools of CD44/CD99-positive cells in glandular tissues and as pluripotent cells located at the pericyte position (*Crisan et al., 2008*; *Kim et al., 2014*). Interestingly, it has been reported that normal fibroblast cell lines contain a subset of pluripotent MSCs, indistinguishable from those derived from the bone marrow (*Denu et al., 2016*) and it is established that normal MSCs lose proliferative and differentiation potential within a few passages when cultured in vitro. This phenomenon has been correlated with a defect in lamin A maturation that leads to cellular senescence (*Bellotti et al., 2016*), which is similar to the terminal differentiation into wound-repair-myofibroblasts resulting from the TGF-beta induction of prostate resident MSCs (*Kim et al., 2014*). HPGS fibroblasts experience an increased rate in apoptosis, with subsequent cell divisions, concomitant with mutant Lamin A accumulation (*Bridger and Kill, 2004*), and chronic DNA damage that lead to premature senescence (*Wheaton et al., 2017*),which could be indirect evidence of early depletion of the MSC pool in these cell lines. (*Figure 10B*).

Related to later differentiation events, the depletion of microvasculature resident MSCs occurring as a result of chronic, deficient wound repair would carry a severe impact to adipogenesis. Cells at the pericyte position have been characterized as the primary source for adipocytes in vivo (*Traktuev et al., 2008*; *Zannettino et al., 2008*). These cells, which are CD44, CD90 double positive, phenocopy the pluripotency of tissue resident mesenchymal stem cells, and are able to differentiate into repair-like myofibroblasts expressing smooth muscle alpha actin (*Merfeld-Clauss et al., 2017*). Coupled with a preadipocyte depletion phenotype, lipodystrophy in HGPS has also been experimentally induced by the introduction of progerin expression into a subset of pre adipogenic cells in mice, which led to fibrosis, senescence, and macrophage infiltration, with the ultimate result of white fat depletion (*Revêchon et al., 2017*).

## The fibroblast as a sentinel of general mesenchymal lineage health

The criticism can be made that it is a stretch to make hypotheses about different mesenchymal stem cell pools based only on gene expression from dermal fibroblasts. This is indeed a limitation that must be considered, but primary samples of other mesenchymal lineage tissues from HGPS patients are not available, and there is evidence of interplay between fibroblasts and the MSC lineage. After birth, the bone marrow mesenchymal stem cell niche is in close synergy with pools of fibroblasts (*LeBleu and Neilson, 2020*; *Soundararajan and Kannan, 2018*), tissue resident mesenchymal stem cells (*El*

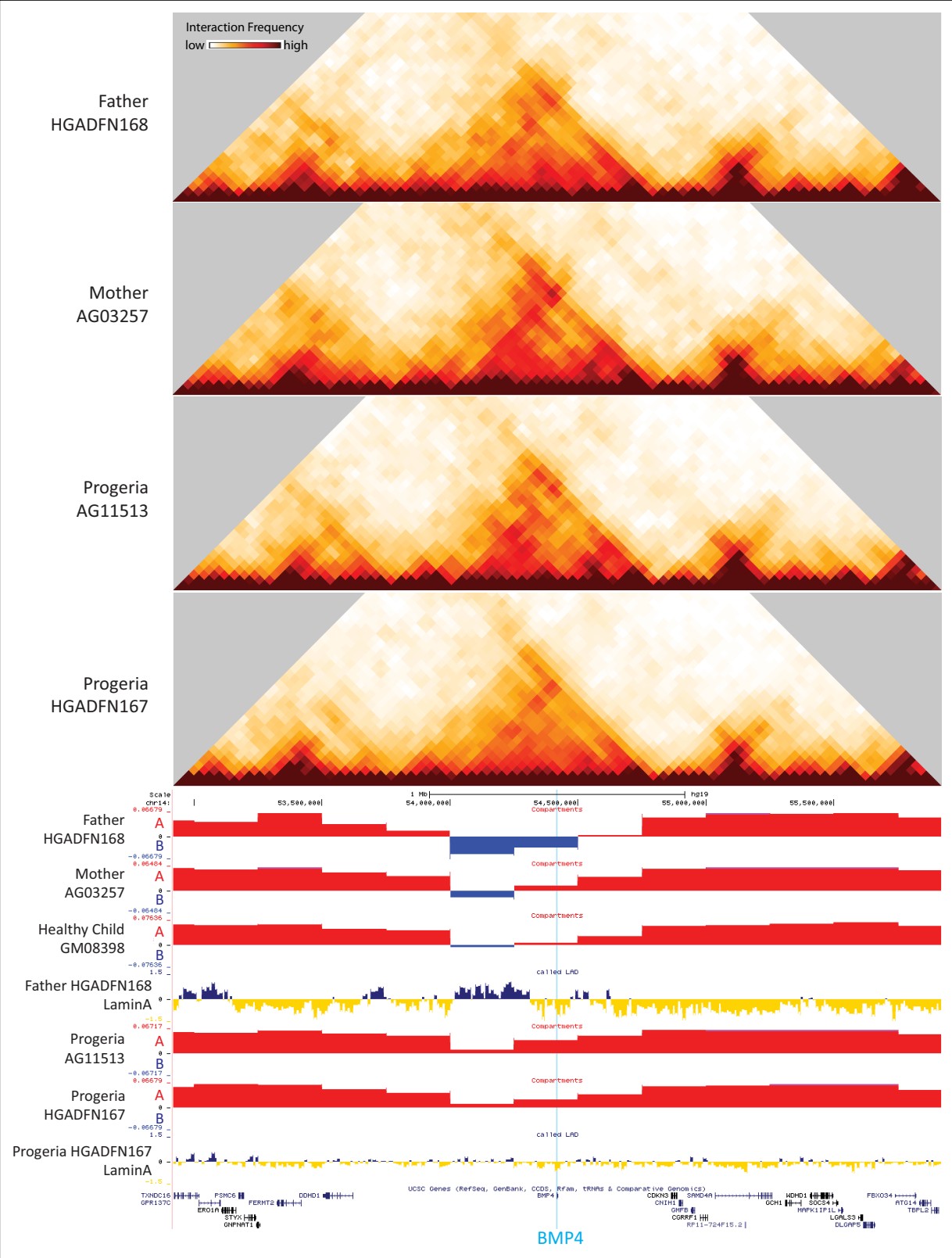

**Figure 9.** 40 kb resolution heatmaps for the parental and HGPS fibroblasts around the BMP4 gene (highlight: blue), aligned to their associated compartment and LAD tracks. The gene is located in a region that shifts toward the A compartment (red) in HGPS compared to WT parent and healthy child controls.

*Figure 9 continued on next page*

*Figure 9 continued*

The online version of this article includes the following figure supplement(s) for figure 9:

**Figure supplement 1.** 40 kb resolution Hi-C contact maps for the parental and HGPS cell lines, aligned with A/B compartment tracks for parental, healthy child, and HGPS cell lines.

**Figure supplement 2.** A/B compartment tracks for parental, healthy child, and HGPS cell lines and Lamin A association tracks for healthy father and HGPS patient centered on genes detected as downregulated in Progeria compared to healthy children.

*Agha et al., 2017*) and pericytes (*Lamagna and Bergers, 2006*). Since these cell pools are in flux, and can compensate for each other under duress (*Di Carlo and Peduto, 2018*; *Direkze et al., 2004*; *Ozerdem et al., 2005*; *San Martin et al., 2014*), comparing the global transcriptional status of fibroblasts between a patient cohort and its aged-matched controls can provide an insight into potential systemic deficits and compensatory mechanisms at play.

## Ideas and speculation

Overall, the cohorts of biological processes that we observe to be misregulated across different age groups of Progeria patients lead us to speculate the impact of this misregulation across a series of events in development, which will warrant further investigation:

1. Biology of priority: the formation of bone 'fires' at specific times. These events are 'non-negotiable', potentially taking precedence over other concomitant MSC fates. Deficits in endochondral ossification can condition the system to draw from pericyte and tissue resident MSCs in an attempt to compensate.
2. The loss of bone fusion and regression of bone length observed in teenaged HGPS patients is an indirect result of deficits in arrest of chondrogenesis proliferation, that occurred in early childhood. These deficits may result in depletion of MSC pools, both in the bone marrow and in tissues
3. Depletion of the pericyte niche is exacerbated by a chronic, incompetent, wound repair response. Ultimately, this depletion will result in loss of microvasculature integrity and subsequent vascular events observed in HGPS, such as vascular stiffness and atherosclerosis.
4. In adipose tissue, the activation of MSCs located at the pericyte niche for addressing deficient wound repair response or osteogenesis, could lead to a deficient differentiation into adipocytes
5. Elaborating on proposed MSC interventions (*Infante and Rodríguez, 2021*) and current gene editing efforts (*Koblan et al., 2021*), targeting the bone marrow niche in young patients could rescue the later HGPS phenotypes.

## Conclusion

Our comprehensive analysis of previously published and newly generated RNA-seq datasets for Progeria fibroblast cells enabled careful age-stratified comparisons. Comparing young children and teenagers with Progeria to age-matched controls, middle-aged, and elderly adults, we find misregulated genes which suggest important defects in cell and tissue repair biology, epigenetics, metabolism (calcium, lipids), and other functions important to the mesenchymal cell lineage. Our comparisons show altered regulation of chondrocyte commitment genes in the fibroblasts of Progeria patients who are at the age of postnatal ossification compared to typically developing age-matched controls. Interestingly, the Progeria gene expression patterns are distinct from older adult fibroblast expression patterns, emphasizing that this disease does not simply mimic normal aging. While genome spatial compartmentalization in these patients is overall weakened, much

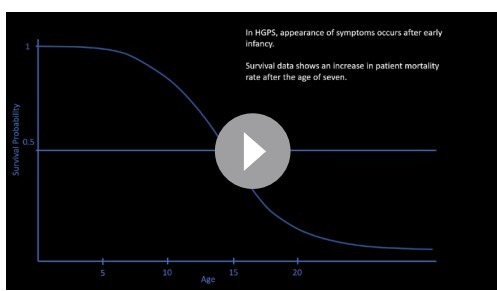

**Video 1.** A visual explanation of ideas presented in the Discussion, hypotheses derived from transcriptomics results.

https://elifesciences.org/articles/81290/figures#video1

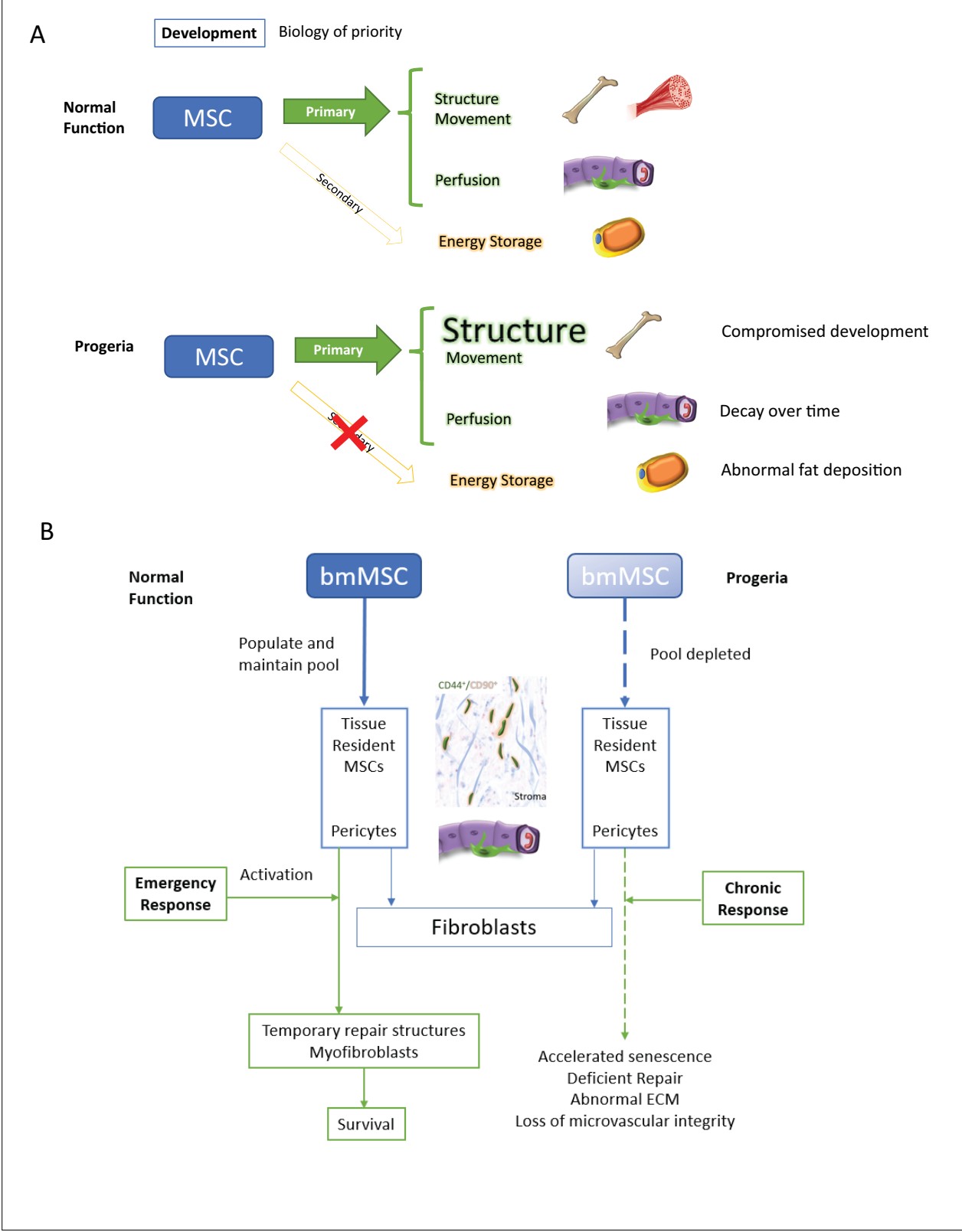

**Figure 10.** Discussion model. HGPS affects differentiation commitment and subsequent biology of priority during early development, which results in premature depletion of MSC pools.

of the gene misregulation occurs in regions that do not change their compartment status or lamin association. However, a few key misregulated genes in the osteogenic and adipogenic lineage show a clear concordant switch in spatial compartmentalization or lamin association, suggesting that some alterations in these pathways may result directly from the influence of progerin on chromosome structure.

# Materials and methods

**Key resources table**

| Reagent type (species) or resource | Designation | Source or reference | Identifiers | Additional information |
|---|---|---|---|---|
| Biological sample (*Homo sapiens*) | Purified RNA | Coriell Institute | AG01178 | Male 20 y/o |
| Biological sample (*Homo sapiens*) | Purified RNA | Coriell Institute | AG03198 | Female 10 y/o |
| Biological sample (*Homo sapiens*) | Purified RNA | Coriell Institute | AG06917 | Male 3 y/o |
| Biological sample (*Homo sapiens*) | Purified RNA | Coriell Institute | AG07493 | Female 2 y/o |
| Biological sample (*Homo sapiens*) | Purified RNA | Coriell Institute | AG08466 | Female 8 y/o |
| Biological sample (*Homo sapiens*) | Purified RNA | Coriell Institute | AG10578 | Male 17 y/o |
| Biological sample (*Homo sapiens*) | Purified RNA | Coriell Institute | AG10677 | Male 4 y/o |
| Biological sample (*Homo sapiens*) | Purified RNA | Coriell Institute | AG11572 | Female 2 y/o |
| Biological sample (*Homo sapiens*) | Purified RNA | Coriell Institute | GM01178 | Male 20 y/o |
| Biological sample (*Homo sapiens*) | Purified RNA | Coriell Institute | GM01972 | Female 14 y/o |
| Biological sample (*Homo sapiens*) | Purified RNA | Coriell Institute | AG11513 | Female 8 y/o |
| Cell line (*Homo sapiens*) | HGPS human primary dermal fibroblast | Progeria Research Foundation (PRF) Cell and Tissue Bank | HGADFN167 | 8-year-old male progeria patient |
| Cell line (*Homo sapiens*) | WT human primary dermal fibroblast | Progeria Research Foundation (PRF) Cell and Tissue Bank | HGADFN168 | Father of progeria patient |
| Cell line (*Homo sapiens*) | HGPS human primary dermal fibroblast | Coriell Institute | AG11513 | 8-year-old female progeria patient |
| Cell line (*Homo sapiens*) | WT human primary dermal fibroblast | Coriell Institute | AG03257 | Mother of progeria patient |
| Cell line (*Homo sapiens*) | WT human primary dermal fibroblast | Coriell Institute | GM08398 | 8-year-old male healthy child |
| Peptide, recombinant protein | HindIII | New England Biolabs | R0104L | |
| Peptide, recombinant protein | DpnII | New England Biolabs | R0543L | |
| Peptide, recombinant protein | T4 DNA Ligase | Invitrogen | 15224041 | |
| Peptide, recombinant protein | DNA Polymerase I Klenow Fragment | New England Biolabs | M0210L | |

*Continued on next page*

*Continued*

| Reagent type (species) or resource | Designation | Source or reference | Identifiers | Additional information |
|---|---|---|---|---|
| Peptide, recombinant protein | T4 DNA Polymerase | New England Biolabs | M0203L | |
| Peptide, recombinant protein | Biotin-dATP | Invitrogen | 19524016 | |
| Commercial assay or kit | Arima-HiC +Kit | Arima Genomics | Mammalian Cell Lines Protocol (A160134 v01) | |
| Commercial assay or kit | NEBNext Ultra II kit | New England Biolabs | E7645S | |
| Commercial assay or kit | NEBNext Multiplex Oligos for Illumina (Index Primers Set 4) | New England Biolabs | E7730S | |
| Commercial assay or kit | NEBNext Multiplex Oligos for Illumina (Index Primers Set 1) | New England Biolabs | E7335S | |
| Software, algorithm | BBDuk | https://github.com/kbaseapps/BBTools | RRID:SCR_016968 | |
| Software, algorithm | STAR aligner | https://github.com/alexdobin/STAR | RRID:SCR_004463 | |
| Software, algorithm | HTSeq-Counts | https://github.com/simon-anders/htseq | RRID:SCR_011867 | |
| Software, algorithm | DESeq2 | https://bioconductor.org/packages/release/bioc/html/DESeq2.html | | |
| Software, algorithm | cMapping | https://github.com/dekkerlab/cMapping (*Lajoie et al., 2015*; *Lajoie and Oomen, 2015*) | | v1.0.6; Bryan Lajoie |
| Software, algorithm | cWorld-dekker | https://github.com/dekkerlab/cworld-dekker (*Lajoie and Venev, 2019*) | | v0.41.1; Bryan Lajoie |
| Software, algorithm | ComBat-seq | https://github.com/zhangyuqing/ComBat-seq (*Zhang et al., 2020a*; *Zhang et al., 2020b*) | | |

## RNA-seq

Ten micrograms of purified RNA from several progeria cell samples were acquired from the Coriell Institute (Camden, NJ) as follows: AG06917 (HGPS. Male 3 y/o), AG10578 (HGPS. Male 17 y/o), AG11572 (HGPS. Female 2 y/o), AG10677 (HGPS. Male 4 y/o), AG08466 (HGPS. Female 8 y/o), AG03198 (HGPS. Female 10 y/o), AG07493 (HGPS. Female 2 y/o), AG01178 (HGPS. Male 20 y/o), AG11513 (HGPS. Female 8 y/o), GM01972 (HGPS. Female 14 y/o), GM01178 (HGPS. Male 20 y/o). After internal quality control upon receipt, RNA-seq library construction and sequencing of two technical replicates was carried out by Genewiz (South Plainfield, NJ). Further, previously published RNA-seq datasets from HPGS fibroblasts and normal controls were included in the study (*Appendix 1—tables 1–3*; *Fleischer et al., 2018*; *Ikegami et al., 2020*; *Köhler et al., 2020*; *Mateos et al., 2018*).

## RNA-seq data processing

The fastq reads were first processed with BBDuk tool (https://github.com/kbaseapps/BBTools), performing adapter trimming with parameters "ktrim = r k=23 mink = 11 hdist = 1". Adapter trimmed reads were processed for quality trimming using the BBDuk tool to discard reads with quality score lower than 28 (parameters "qtrim = r trimq = 28"). Following the adapter and quality trimming steps, the reads were aligned to the reference genome hg19 using STAR aligner (https://github.com/alexdobin/STAR) with both '--outFilterScoreMinOverLread' and ' --outFilterMatchNminOverLread' parameters set to 0.2. Finally, the mapped reads were sorted based on genomic coordinates and feature count was performed with HTSeq-Counts (https://github.com/simon-anders/htseq).

## Batch effect removal

Since the RNA-seq files used in this study were generated in different laboratories, using different technologies, the raw gene counts produced by HTSeq-Counts suffer from batch effect. To mitigate

that issue, the raw counts are batch effect adjusted using the ComBat-seq tool (https://github.com/zhangyuqing/ComBat-seq). For this purpose, files from the same laboratory are assigned the same batch number. In addition, the sex and status of the samples are provided as the biological covariates to the ComBat-seq tool to preserve that signal in the adjusted data.

## Differential expression analysis

Differential gene expression analysis was performed between different groups of diseased and healthy samples using DESeq2 (https://bioconductor.org/packages/release/bioc/html/DESeq2.html) tool. HGPS samples from Young (0–7 y/o) or teenage (13+y/o) patients were compared to age matched (AM), middle aged (M), and old (O) control samples (*Table 1—source data 1*). Further, to correlate gene expression profiles with known patient survival statistics, comparisons were further refined by stratifying early infancy progeria patients (0–3 years old) and older children (4–7 years old), comparing those populations to age matched, middle aged and old control samples, as before (*Table 2—source data 1*).

## Gene ontology analysis

Genes defined as up/down regulated for each of the age group comparisons, were considered for gene ontology analysis based on an FDR adjusted p-value cutoff of 0.001. These gene lists, in turn, were then analyzed using Metascape (*Zhou et al., 2019*). Metascape output files were manually clustered into gene ontology themes, observed throughout the analysis. Gene lists derived from this manual clustering were then visualized via heatmaps to facilitate comparisons among age groups.

## Gene expression heatmap plotting

To plot expression of specific genes across different samples as a heatmap, the batch adjusted counts for all the genes for each sample were log2 normalized with a pseudo count of 1. The expression of specific genes across specific samples was then extracted and Z-score normalization was performed for each of the genes. Finally, the Z-score normalized values were plotted using the Seaborn python package as a heatmap. (https://seaborn.pydata.org/).

## DamID-seq data processing

All the DamID-seq data was processed as previously reported (*Leemans et al., 2019*), with modifications in the trimming step. Briefly, the bwa mem (https://github.com/lh3/bwa, *Li, 2022*) tool was used to map gDNA reads starting with GATC to a combination of hg19 reference genome with a ribosomal model. For further processing, only the mapped reads having a mapping quality of at least 10 were considered as GATC fragments. Next, the reads were combined into the bins of 40 kb resolution depending on the middle of the GATC fragments and then scaled to 1 M reads. For normalization, log2-ratio of the scaled target over the scaled Dam-only bins was calculated with a pseudo count of 1.

## Cell culture

Human primary dermal fibroblast cell lines were obtained from The Progeria Research Foundation (PRF) Cell and Tissue Bank (HGPS: HGADFN167 and healthy adult: HGADFN168). Dermal fibroblasts (HGPS: AG11513, healthy adult: AG03257, and healthy child GM08398) were purchased from Coriell Institute (Camden, NJ). Cells were grown in DMEM (Gibco) supplemented with 15% FBS (Corning), 1% Pen-strep (Gibco), and 1% L-glutamine (Gibco). Cells were passaged at a density of 80%.

## Chromosome conformation capture (Hi-C)

Hi-C experiments were performed according to standard protocols (*Golloshi et al., 2018*). The starting material was comprised of skin fibroblasts of parents of HGPS patients (Mother AG03257, Father HGADFN168), a healthy age matched child (GM08398) and two HGPS patient fibroblast samples (HGADFN167 and AG11513). Further, HGADFN168 and HGADFN167, belonging to a parent-child matched cells were analyzed both at an early and late passage: p12-27 and p12-19, respectively.

Briefly, ~5 million cells were fixed with 1% formaldehyde, suspended in cell lysis buffer for permeabilization, and homogenized by douncing. Crosslinked chromatin was digested overnight with HindIII (HGADFN167 and HGADFN168) or DpnII (HGADFN167, HGADFN168, AG03257 and AG11513). GM08398 cells were processed using the Arima-HiC +Kit from Arima Genomics following the protocol

for Mammalian Cell Lines (A160134 v01). Sticky ends were filled in with biotin-dATP (Invitrogen), and the blunt ends of interacting fragments were ligated together. DNA was purified by two phenol-chloroform extractions and ethanol precipitation. Biotin-dATP at unligated ends was removed, and the DNA was sheared to a target size of 200–400 bp by a Covaris sonicator (Covaris, M220). DNA between 100–400 bp was selected for using AMPure XP beads (Beckman Coulter). Biotinylated DNA was pulled down using streptavidin coated magnetic beads and prepared for multiplex sequencing on an Illumina platform using the NEBNext Ultra II kit (NEB, E7645S). All end preparation, adaptor ligation, and PCR amplification steps were carried out on bead bound DNA libraries. Sequencing was carried out on Illumina HiSeq 3000 or NovaSeq platforms with 75 bp or 150 bp paired end reads. All Hi-C data statistics are presented in *Appendix 1—table 4*.

## Analysis of Hi-C data

Sequencing reads were mapped to the reference human genome hg19, filtered, and iteratively corrected using previously published pipelines (*Imakaev et al., 2012*), available on github (https://github.com/dekkerlab/cMapping). Publicly available tools (https://github.com/dekkerlab/cworld-dekker) were used to produce Hi-C heatmaps at 2.5 Mb and 250 kb resolution and to perform principal component analysis (using the matrix2compartment script) to generate compartment tracks at 250 kb resolution, assigning A and B compartments to positive and negative PC1 values, respectively. Values from replicate experiments were averaged, by bin, to produce the final compartment track. TAD boundary strength analysis was carried out using the insulation score approach (*Crane et al., 2015*) (matrix2insulation) from this cworld package with 500 kb insulation square using 40 kb resolution contact maps.

## ChIP-seq data and processing: Lamin associated domain analysis

Raw LMNA ChIP-seq data was obtained from the NCBI Gene Expression Omnibus (GEO) under accession number GSE41764 (*McCord et al., 2013*). For comparison, DamID-seq for the fibroblast line HFFc6 (van Steensel Lab, Netherlands Cancer Institute) was obtained from the 4D Nucleome data portal (*Dekker et al., 2017*; *Reiff et al., 2021*), through the bio sample identifier 4DNBSR7TC87A.

The fastq reads were first processed for adapter and quality trimming with the help of BBDuk tool. The reads were then aligned to the hg19 reference genome using STAR aligner. Finally, both the target and input mapped reads were binned at 40 kb resolution and log2-ratio of the target over the input bins was calculated using the 'bamCompare' function of deepTools with parameters "`--operation log2 -bs 40000 --ignoreDuplicates --minMappingQuality 30 --scaleFactorsMethod SES --effectiveGenomeSize 2864785220`".

## Acknowledgements

The authors thank Alvaro Rodriguez Gonzalez for his assistance with Hi-C data processing and Enrique Pacheco San Martin for his assistance with graphics and media design. The concept of the biology of priority during development and repair was adapted from discussion with, and coursework designed by, David Rowley Ph.D. (Baylor College of Medicine). This work was supported by NIH NIGMS Grant R35GM133557 to RPM. R San Martin was supported by a postdoctoral fellowship from the American Cancer Society (134060-PF-19-183-01-CSM).

## Additional information

### Funding

| Funder | Grant reference number | Author |
| --- | --- | --- |
| National Institute of General Medical Sciences | R35GM133557 | Rachel Patton McCord |
| American Cancer Society | 134060-PF-19-183-01-CSM | Rebeca San Martin |

The funders had no role in study design, data collection and interpretation, or the decision to submit the work for publication.

## Author contributions
Rebeca San Martin, Conceptualization, Formal analysis, Visualization, Writing – original draft, Writing – review and editing; Priyojit Das, Writing – review and editing, Investigation, Conceptualization, Writing – original draft; Jacob T Sanders, Conceptualization, Data curation, Formal analysis, Investigation, Writing – original draft, Writing – review and editing; Ashtyn M Hill, Investigation, Data curation; Rachel Patton McCord, Conceptualization, Data curation, Formal analysis, Funding acquisition, Investigation, Project administration, Supervision, Visualization, Writing – original draft, Writing – review and editing

## Author ORCIDs
Rebeca San Martin ⓘ http://orcid.org/0000-0001-7249-3922
Priyojit Das ⓘ http://orcid.org/0000-0002-6774-6718
Rachel Patton McCord ⓘ http://orcid.org/0000-0003-0010-5323

## Decision letter and Author response
Decision letter https://doi.org/10.7554/eLife.81290.sa1
Author response https://doi.org/10.7554/eLife.81290.sa2

---

# Additional files

## Supplementary files
• MDAR checklist

## Data availability
All RNA-seq and Hi-C data contributed by this study is available on GEO at GSE206707 (https://www.ncbi.nlm.nih.gov/geo/query/acc.cgi?acc=GSE206707).

The following dataset was generated:

| Author(s) | Year | Dataset title | Dataset URL | Database and Identifier |
|---|---|---|---|---|
| San Martin R, Das P, Sanders JT, Hill AM, McCord RP | 2022 | Transcriptional profiling of Hutchinson-Gilford Progeria syndrome fibroblasts reveals deficits in mesenchymal stem cell commitment to differentiation related to early events in endochondral ossification | https://www.ncbi.nlm.nih.gov/geo/query/acc.cgi?acc=GSE206707 | NCBI Gene Expression Omnibus, GSE206707 |

The following previously published datasets were used:

| Author(s) | Year | Dataset title | Dataset URL | Database and Identifier |
|---|---|---|---|---|
| Lyko F, Rodriguez-Paredes M | 2020 | Epigenetic deregulation of lamina-associated domains in Hutchinson-Gilford Progeria Syndrome (RNA-Seq) | https://www.ncbi.nlm.nih.gov/geo/query/acc.cgi?acc=GSE150137 | NCBI Gene Expression Omnibus, GSE150137 |
| Fleischer JG, Schulte R, Tsai H, Tyagi S, Ibarra A, Shokhirev MN, Huang L, Hetzer MW, Navlakha S | 2018 | Predicting age from the transcriptome of human dermal fibroblasts | https://www.ncbi.nlm.nih.gov/geo/query/acc.cgi?acc=GSE113957 | NCBI Gene Expression Omnibus, GSE113957 |
| Ikegami K, Secchia S | 2020 | RNA-seq reported in "Phosphorylated Lamin A/C in the nuclear interior binds active enhancers associated with abnormal transcription in progeria" | https://www.ncbi.nlm.nih.gov/geo/query/acc.cgi?acc=GSE113343 | NCBI Gene Expression Omnibus, GSE113343 |

*Continued on next page*

*Continued*

| Author(s) | Year | Dataset title | Dataset URL | Database and Identifier |
|---|---|---|---|---|
| Mateos J, Fafián-Labora J, Morente-López M, Lesende-Rodriguez I, Monserrat L, Ódena MA, Oliveira E, de Toro J, Arufe MC | 2018 | Quantitative whole transcriptomics sequencing of progeria-derived cells point to a key role of nucleotide metabolism in premature aging | https://www.ncbi.nlm.nih.gov/geo/query/acc.cgi?acc=GSE113648 | NCBI Gene Expression Omnibus, GSE113648 |

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

## Appendix 1

This appendix includes details on the patients and experimental approaches and statistics for the RNA-seq data and Hi-C data included in this study.

**Appendix 1—table 1.** RNA-seq datasets from progeria patients used in this study.

| Progeria Patient | Age | Sex | Race | RNA-seq approach | Lab | GEO Series |
|---|---|---|---|---|---|---|
| HGADFN155 | 1 yr 2 mo | F | Unknown | polyA (TruSeq RNA Sample Preparation v2 protocol) | Rodríguez-Paredes | GSE150137 |
| NG07493 | 2 yr | F | White | rRNA depletion | McCord | GSE206684 |
| NG11572 | 2 yr | F | White | rRNA depletion | McCord | GSE206684 |
| HGADFN188 | 2 yr 3 mo | F | Unknown | polyA (TruSeq Stranded mRNA) | Fleischer | GSE113957 |
| HGADFN188 | 2 yr 3 mo | F | Unknown | polyA (TruSeq RNA Sample Preparation v2 protocol) | Rodríguez-Paredes | GSE150137 |
| HGADFN367 | 3 yr | F | Unknown | polyA (TruSeq Stranded mRNA) | Fleischer | GSE113957 |
| NG06917 | 3 yr | M | White | rRNA depletion | McCord | GSE206684 |
| HGADFN127 | 3 yr 9 mo | F | Unknown | polyA (TruSeq Stranded mRNA) | Fleischer | GSE113957 |
| NG10677 | 4 yr | M | White | rRNA depletion | McCord | GSE206684 |
| HGADFN164 | 4 yr 8 mo | F | Unknown | polyA (TruSeq Stranded mRNA) | Fleischer | GSE113957 |
| HGADFN164 | 4 yr 8 mo | F | Unknown | polyA (TruSeq RNA Sample Preparation v2 protocol) | Rodríguez-Paredes | GSE150137 |
| HGADFN122 | 5 yr | F | Unknown | polyA (TruSeq Stranded mRNA) | Fleischer | GSE113957 |
| HGADFN178 | 6 yr 11 mo | F | Unknown | polyA (TruSeq Stranded mRNA) | Fleischer | GSE113957 |
| AG11513 | 8 yr | F | White | polyA (TruSeq Stranded mRNA) | Fleischer | GSE113957 |
| NG08466 | 8 yr | F | White | rRNA depletion | McCord | GSE206684 |
| NG11513 | 8 yr | F | White | rRNA depletion | McCord | GSE206684 |
| HGADFN167 | 8 yr 5 mo | M | Unknown | polyA (NEBNext Ultra DNA Library Prep Kit) | Ikegami | GSE113343 |
| HGADFN167-2 | 8 yr 5 mo | M | Unknown | polyA (NEBNext Ultra DNA Library Prep Kit) | Ikegami | GSE113343 |
| HGADFN167 | 8 yr 5 mo | M | Unknown | polyA (TruSeq Stranded mRNA) | Fleischer | GSE113957 |
| HGADFN167 | 8 yr 5 mo | M | Unknown | polyA (TruSeq RNA Sample Preparation v2 protocol) | Rodríguez-Paredes | GSE150137 |
| HGADFN169 | 8 yr 6 mo | M | Unknown | polyA (TruSeq Stranded mRNA) | Fleischer | GSE113957 |
| HGADFN169 | 8 yr 6 mo | M | Unknown | polyA (TruSeq RNA Sample Preparation v2 protocol) | Rodríguez-Paredes | GSE150137 |
| HGADFN143 | 8 yr 10 mo | M | Unknown | polyA (TruSeq Stranded mRNA) | Fleischer | GSE113957 |

*Appendix 1—table 1 Continued on next page*

Appendix 1—table 1 Continued

| Progeria Patient | Age | Sex | Race | RNA-seq approach | Lab | GEO Series |
|---|---|---|---|---|---|---|
| HGADFN143 | 8 yr 10 mo | M | Unknown | polyA (TruSeq RNA Sample Preparation v2 protocol) | Rodríguez-Paredes | GSE150137 |
| AG03199 | 10 yr | F | White | polyA (Illumina SureSelect Strand Specific RNA library Prep) | Arufe | GSE113648 |
| NG03198 | 10 yr | F | White | rRNA depletion | McCord | GSE206684 |
| AG03513 | 13 yr | M | White Mexican | polyA (Illumina SureSelect Strand Specific RNA library Prep) | Arufe | GSE113648 |
| AG11498 | 14 yr | M | African American | polyA (NEBNext Ultra DNA Library Prep Kit) | Ikegami | GSE113343 |
| AG11498-2 | 14 yr | M | African American | polyA (NEBNext Ultra DNA Library Prep Kit) | Ikegami | GSE113343 |
| NA01972 | 14 yr | F | White | rRNA depletion | McCord | GSE206684 |
| NG10578 | 17 yr | M | White | rRNA depletion | McCord | GSE206684 |
| NA01178 | 20 yr | M | Unknown | rRNA depletion | McCord | GSE206684 |
| NG01178 | 20 yr | M | Unknown | rRNA depletion | McCord | GSE206684 |

**Appendix 1—table 2.** RNA-seq datasets from adult controls used in this study.

| Donor | Age | Sex | Race | RNA-seq approach | Lab | GEO Series |
|---|---|---|---|---|---|---|
| *Parent of Progeria Patient* | | | | | | |
| AG03257 | 35 | F | | polyA (Illumina SureSelect Strand Specific RNA library Prep) | Arufe | GSE113648 |
| AG03512 | 41 | F | | polyA (Illumina SureSelect Strand Specific RNA library Prep) | Arufe | GSE113648 |
| *Healthy Mid-Age Adult* | | | | | | |
| AG07124 | 26 | F | White | polyA (TruSeq Stranded mRNA) | Fleischer | GSE113957 |
| GM00495 | 29 | M | Unknown | polyA (TruSeq Stranded mRNA) | Fleischer | GSE113957 |
| AG07478 | 29 | M | White | polyA (TruSeq Stranded mRNA) | Fleischer | GSE113957 |
| AG04054 | 29 | M | White | polyA (TruSeq Stranded mRNA) | Fleischer | GSE113957 |
| AG09599 | 30 | F | White | polyA (TruSeq Stranded mRNA) | Fleischer | GSE113957 |
| AG09605 | 30 | M | White | polyA (TruSeq Stranded mRNA) | Fleischer | GSE113957 |
| GM04503 | 31 | F | White | polyA (TruSeq Stranded mRNA) | Fleischer | GSE113957 |
| GM04504 | 31 | F | White | polyA (TruSeq Stranded mRNA) | Fleischer | GSE113957 |
| GM00043 | 32 | F | Black, Puerto Rican | polyA (TruSeq Stranded mRNA) | Fleischer | GSE113957 |
| GM01650 | 37 | F | Unknown | polyA (TruSeq Stranded mRNA) | Fleischer | GSE113957 |
| GM01717 | 39 | M | White | polyA (TruSeq Stranded mRNA) | Fleischer | GSE113957 |
| AG16358 | 41 | F | White | polyA (TruSeq Stranded mRNA) | Fleischer | GSE113957 |
| AG13967 | 41 | M | White | polyA (TruSeq Stranded mRNA) | Fleischer | GSE113957 |
| AG04063 | 43 | M | White | polyA (TruSeq Stranded mRNA) | Fleischer | GSE113957 |

*Older Adult Control*

| | | | | | | |
|---|---|---|---|---|---|---|
| GM03525 | 80 | F | White | polyA (TruSeq Stranded mRNA) | Fleischer | GSE113957 |
| GM01706 | 82 | F | White | polyA (TruSeq Stranded mRNA) | Fleischer | GSE113957 |
| AG04386 | 83 | M | White | polyA (TruSeq Stranded mRNA) | Fleischer | GSE113957 |
| AG11744 | 84 | F | White | polyA (TruSeq Stranded mRNA) | Fleischer | GSE113957 |
| AG11725 | 84 | F | White | polyA (TruSeq Stranded mRNA) | Fleischer | GSE113957 |
| AG05274 | 84 | M | White | polyA (TruSeq Stranded mRNA) | Fleischer | GSE113957 |
| AG05247 | 87 | F | White | polyA (TruSeq Stranded mRNA) | Fleischer | GSE113957 |
| AG04662 | 87 | M | White | polyA (TruSeq Stranded mRNA) | Fleischer | GSE113957 |
| AG13129 | 89 | M | White | polyA (TruSeq Stranded mRNA) | Fleischer | GSE113957 |
| AG12788 | 90 | M | White | polyA (TruSeq Stranded mRNA) | Fleischer | GSE113957 |
| AG07725 | 91 | M | White | polyA (TruSeq Stranded mRNA) | Fleischer | GSE113957 |
| AG09602 | 92 | F | White | polyA (TruSeq Stranded mRNA) | Fleischer | GSE113957 |
| AG04064 | 92 | M | White | polyA (TruSeq Stranded mRNA) | Fleischer | GSE113957 |
| AG08433 | 94 | M | White | polyA (TruSeq Stranded mRNA) | Fleischer | GSE113957 |
| AG04059 | 96 | M | White | polyA (TruSeq Stranded mRNA) | Fleischer | GSE113957 |

**Appendix 1—table 3.** RNA-seq datasets from children controls used in this study.

| Healthy Child | Age | Sex | Race | RNA-seq approach | Lab | GEO Series |
|---|---|---|---|---|---|---|
| AG08498 | 1 | M | Asian | polyA (TruSeq Stranded mRNA) | Fleischer | GSE113957 |
| GM00969 | 2 | F | Caucasian | polyA (TruSeq Stranded mRNA) | Fleischer | GSE113957 |
| GM00969 | 2 | F | Caucasian | polyA (TruSeq RNA Sample Preparation v2 protocol) | Rodríguez-Paredes | GSE150137 |
| GM05565 | 3 | M | Latino/Hispanic | polyA (TruSeq Stranded mRNA) | Fleischer | GSE113957 |
| GM00498 | 3 | M | Unknown | polyA (TruSeq Stranded mRNA) | Fleischer | GSE113957 |
| GM05400 | 6 | M | Black | polyA (TruSeq Stranded mRNA) | Fleischer | GSE113957 |
| GM00409 | 7 | M | Caucasian | polyA (TruSeq Stranded mRNA) | Fleischer | GSE113957 |
| GM00499 | 8 | M | Caucasian | polyA (TruSeq Stranded mRNA) | Fleischer | GSE113957 |
| GM08398 | 8 | M | Caucasian | polyA (NEBNext Ultra DNA Library Prep Kit) | Ikegami | GSE113343 |
| GM08398-2 | 8 | M | Caucasian | polyA (NEBNext Ultra DNA Library Prep Kit) | Ikegami | GSE113343 |
| GM08398 | 8 | M | Caucasian | polyA (TruSeq Stranded mRNA) | Fleischer | GSE113957 |
| GM00038 | 9 | F | Black | polyA (TruSeq Stranded mRNA) | Fleischer | GSE113957 |
| GM01652 | 11 | F | Caucasian | polyA (TruSeq Stranded mRNA) | Fleischer | GSE113957 |
| GM01582 | 11 | F | Caucasian | polyA (TruSeq Stranded mRNA) | Fleischer | GSE113957 |

*Appendix 1—table 3 Continued on next page*

*Appendix 1—table 3 Continued*

| Healthy Child | Age | Sex | Race | RNA-seq approach | Lab | GEO Series |
|---|---|---|---|---|---|---|
| AG16409 | 12 | M | Caucasian | polyA (TruSeq Stranded mRNA) | Fleischer | GSE113957 |
| GM07532 | 16 | F | Unknown | polyA (TruSeq Stranded mRNA) | Fleischer | GSE113957 |
| GM07753 | 17 | M | Caucasian | polyA (TruSeq Stranded mRNA) | Fleischer | GSE113957 |
| GM07492 | 17 | M | Caucasian | polyA (NEBNext Ultra DNA Library Prep Kit) | Ikegami | GSE113343 |
| GM07492-2 | 17 | M | Caucasian | polyA (NEBNext Ultra DNA Library Prep Kit) | Ikegami | GSE113343 |
| GM07492 | 17 | M | Caucasian | polyA (TruSeq Stranded mRNA) | Fleischer | GSE113957 |
| GM08399 | 19 | F | Caucasian | polyA (TruSeq Stranded mRNA) | Fleischer | GSE113957 |

**Appendix 1—table 4.** Hi-C Data Statistics.

| Sample Name | Enzyme | Genotype | Gender, Age | Raw Reads | Both Sides Mapped | % Map | % Dangling Ends | Valid Pairs | Unique Valid Pairs | %Cis |
|---|---|---|---|---|---|---|---|---|---|---|
| Progeria-167-DpnII-P19-R1 | DpnII | HGPS | Male, 8Y | 364,961,495 | 220,942,665 | 60.5 | 3.05 | 208,963,277 | 152,974,533 | 54.69 |
| Progeria-168-DpnII-P16-R1 | DpnII | WT | Male, 40Y | 358,703,556 | 220,880,406 | 61.6 | 3.48 | 212,152,596 | 148,022,234 | 54.98 |
| Progeria-AG03257-P7-R1 | DpnII | WT | Female, 35Y | 274,364,560 | 177,877,476 | 64.8 | 0.95 | 174,730,268 | 132,694,872 | 49.65 |
| Progeria-AG03257-P7-R3 | DpnII | WT | Female, 35Y | 127,656,906 | 73,190,497 | 57.3 | 2.26 | 70,541,316 | 46,463,429 | 83.56 |
| Progeria-AG11513-P7-R1 | DpnII | HGPS | Female, 8Y | 251,483,696 | 154,772,051 | 61.5 | 0.46 | 153,219,873 | 116,077,671 | 48.77 |
| Progeria-AG11513-P7-R2 | DpnII | HGPS | Female, 8Y | 251,776,055 | 153,160,225 | 60.8 | 0.75 | 151,305,617 | 114,631,955 | 50.22 |
| Progeria-HGADFN167-P12-R1 | HindIII | HGPS | Male, 8Y | 165,040,682 | 116,708,094 | 70.7 | 24.91 | 86,987,628 | 81,047,934 | 67.32 |
| Progeria-HGADFN167-P19-R1 | HindIII | HGPS | Male, 8Y | 202,090,523 | 136,172,100 | 67.4 | 14.12 | 116,222,620 | 95,537,352 | 82.72 |
| Progeria-HGADFN168-P12-R1 | HindIII | WT | Male, 40Y | 194,414,002 | 136,139,915 | 70.0 | 16.61 | 112,990,317 | 77,889,695 | 81.23 |
| Progeria-HGADFN168-P27-R1 | HindIII | WT | Male, 40Y | 157,637,031 | 110,558,519 | 70.1 | 15.06 | 93,504,712 | 86,842,660 | 67.32 |
| Progeria-GM08398-P13-R1 | Arima | WT | Male, 8Y | 475,221,692 | 309,944,409 | 65.2 | 1.59 | 301,485,345 | 182,322,975 | 83.07 |

