## [Editor Report]

This manuscript is of interest to researchers investigating genetic mechanisms of aging and transcriptional regulation of developmental processes in mesenchyme-derived tissues. In this study, fibroblast cell lines from patients with and without Hutchinson-Gilford Progeria were compared to pinpoint the molecular mechanisms leading to the phenotypes of persons with this condition. The identification of five major dysregulated functional hubs in fibroblast cell lines derived from Hutchinson-Gilford Progeria Syndrome (HGPS) patients provides a unique opportunity for others working on this disorder to utilize animal models to validate the authors' hypotheses.

---

## [Decision Letter]

**Decision letter after peer review:**

Thank you for submitting your article "Transcriptional profiling of Hutchinson-Gilford Progeria syndrome fibroblasts reveals deficits in mesenchymal stem cell commitment to differentiation related to early events in endochondral ossification" for consideration by *eLife*. Your article has been reviewed by 2 peer reviewers, one of whom is a member of our Board of Reviewing Editors, and the evaluation has been overseen by Carlos Isales as the Senior Editor. The reviewers have opted to remain anonymous.

Essential revisions:

1. There is no conclusion to this paper and one needs to be added. The speculations at the end of the discussion are interesting but not tied into the rest of the narrative. The discussion as it exists is a series of speculations built on preceding speculations, all derived from previously published studies. While this reviewer appreciates the attempt to fit this study's findings into a current paradigm that hypothesizes stem cell depletion as a pathomechanism in HGPS, there is no direct evidence to support that hypothesis in this study, especially as it applies to endochondral ossification. Although it is beyond the scope of the current study, one potential approach to address this would be to experimentally quantitate MSCs in different tissue compartments at multiple timepoints during an HGPS animal model's lifespan.

2. The abstract is just a series of “what was done” statements and is not effective at highlighting the main findings and take-home conclusion.

3. The sample set needs to be more completely described up front in this paper (what are the patient demographics of the final data set). This is available to a certain extent but you really need to dig for it. Further, it needs to be clearly stated that these cells were never differentiated into mesenchymal stem cells.

4. The cartoon at the top of the figuring to denote the focus of that figure is less effective than just stating in words in a title what that figure is about.

5. The father-son data is interesting but this is an n of one pair and needs to be replicated to support the conclusions made. Barring the availability of additional data, this section needs to be either removed or less importance given to these results.

6. Tables 1 and 2 are missing from the manuscript and must be included in the future version.

7. The authors state that their transcriptional analysis reveals eight clusters of altered biological activity in HGPS, but it is not entirely clear what those eight specific clusters are based on the way the section is written.

8. Figure 2C: why was the age-matched control comparison throughout this manuscript excluded from this particular analysis?

9. The 10th paragraph of the Results, pertaining to epigenetic modifications, seems more suited to be presented earlier in the section where the authors present their GO analysis highlighting epigenetic programming and DNA maintenance (3rd paragraph of Results) with regards to Figure 1.

10. 11th paragraph: The underlying theme of the manuscript is that defective mesenchymal stem cell commitment in HGPS results in abnormal development of bone. As such, the gene expression data related to ossification and calcium homeostasis should be provided. Furthermore, BMP4 stated to be upregulated, is not included in the heat plot associated with this finding (Figure 7A).

11. Downregulated genes are shown to be associated with alterations in compartment identity or lamin interactions. There is no mention if similar alterations occur for upregulated genes, which should be included.

12. Some important findings in this study include the dysregulation of pathways involved in blood vessel morphogenesis and maturation, muscle development, and lipid metabolism. Vascularization is essential for normal bone development and tissue, muscle provides mechanical and metabolic influences on bone homeostasis, while lipid metabolism is critical in determining the phenotypic fate of common progenitors of osteoblasts and adipocytes. The authors should therefore comment on the potential roles dysregulation of these pathways has in the HGPS bone phenotype as an alternative to suggesting MSC depletion as the sole driving force behind HGPS pathology.

13. A major shortcoming of this work is the drawing of conclusions on pathomechanisms of HGPS in multiple mesenchyme-derived tissues based on fibroblast transcriptional and epigenetic profiles which are, however, acknowledged by the authors. Specifically, changes in gene expression are attributed to alterations in compartment identity or lamina association in fibroblasts, but no data is provided to suggest those same changes occur identically in other cells or tissues where conclusions are being made. This requires further discussion in the paper.

14. The manuscript would be strengthened by reorganizing the Results section to more clearly present the data pertaining to each of the five functional hubs that were found to be affected in HGPS cell lines in a linear manner.

---

## [Author Response]

Essential revisions:1. There is no conclusion to this paper and one needs to be added. The speculations at the end of the discussion are interesting but not tied into the rest of the narrative. The discussion as it exists is a series of speculations built on preceding speculations, all derived from previously published studies. While this reviewer appreciates the attempt to fit this study's findings into a current paradigm that hypothesizes stem cell depletion as a pathomechanism in HGPS, there is no direct evidence to support that hypothesis in this study, especially as it applies to endochondral ossification. Although it is beyond the scope of the current study, one potential approach to address this would be to experimentally quantitate MSCs in different tissue compartments at multiple timepoints during an HGPS animal model's lifespan.

The first section of the Discussion describes our conclusions that can be drawn directly from the data and analyses that we have presented in this study. To clarify and reiterate these more direct conclusions, we have now included a brief final “Conclusion” paragraph with a heading at the very end of the paper, after the ideas and speculation.

We are glad that the reviewers see value in our efforts to place our results in the context of broader hypotheses about Progeria pathogenesis. Our study builds on the clinical observations of deficits in endochondral ossification in progeria patients, seen in the regression of terminal phalanges and lack of fusion in the clavicle in adolescent patients. Our results provide insights into potential misregulated targets related to these events which we find correlate with the specific ages at which these developmental processes occur. We agree that these results do not provide specific evidence that these genes underlie the deficits seen in patients and that more work is needed to test that idea. We agree that animal studies are beyond the scope of the current study, and we hope that the timely publication of the present analysis will inform the field, and other progeria researchers with access to those animal models, to conduct those types of experiments. We have taken care in the revision to clarify wherever possible what we can conclude directly from our data vs. what is a hypothesis for the future.

2. The abstract is just a series of “what was done” statements and is not effective at highlighting the main findings and take-home conclusion.

We have now revised the abstract to highlight the findings more clearly.

3. The sample set needs to be more completely described up front in this paper (what are the patient demographics of the final data set). This is available to a certain extent but you really need to dig for it. Further, it needs to be clearly stated that these cells were never differentiated into mesenchymal stem cells.

We have modified our RNA-seq sample tables to more clearly present the demographics of all patients and controls included in the final datasets, including both the new RNA-seq data we generated and previously published datasets (Appendix 1 – Tables 1-3). These tables are referenced in the Methods section. We have also included more of this information directly in the manuscript as shown below:

“In this study, we conduct a comprehensive analysis of previously published RNA-seq datasets for HPGS fibroblast lines harboring the typical C>T mutation in the LMNA gene and provide transcriptomics data for nine previously unreported cell lines. By comparing the transcriptional profile of HPGS fibroblasts (33 datasets, 21 patients, 1-20 years old) with fibroblasts derived from age-matched controls (21 datasets, 16 donors, 1-19 years old), healthy adults, (16 donors, 26-43 years old) and healthy old adults (15 donors, 80-96 years old), we provide insight into important defects in repair biology, metabolism (calcium, lipid), and other areas of mesenchymal cell lineage importance.”

Changes have been made to the manuscript to clarify that these fibroblasts were not induced to any of the lineages as follows:

“In this analysis of fibroblasts, we also identified misregulation of gene targets typically associated with the biology of mesenchymal tissue. Specifically, we see misregulation of transcription of genes involved in bone (165 genes. Figure 3),”

We have revised text in other places to remind the reader that our conclusions are drawn from fibroblasts rather than other mentioned cell types, such as in the introduction:

“Our results show that transcription of genes involved in negative regulation of chondrocyte commitment are compromised in fibroblasts from patients that are at the age of onset of postnatal endochondral ossification.”

4. The cartoon at the top of the figuring to denote the focus of that figure is less effective than just stating in words in a title what that figure is about.

We have now added titles to all figures for further clarity, though we also still include the cartoons to cater to more visual readers.

5. The father-son data is interesting but this is an n of one pair and needs to be replicated to support the conclusions made. Barring the availability of additional data, this section needs to be either removed or less importance given to these results.

We attempted to collect an additional parent-child matched pair-- the mother AG03257 sample we have is matched to a Progeria patient fibroblast line AG0199, but this particular Progeria patient’s cells enter senescence so quickly in culture that we could not collect enough cells for a Hi-C experiment. Generally, matched pairs are very limited in their availability. We include the conclusions we present for comparison to previous work (McCord et al., 2013) but we have toned down the language so as not to emphasize the “matched pair” nature of the comparison too much.

6. Tables 1 and 2 are missing from the manuscript and must be included in the future version.

We apologize for this oversight and have now included the tables.

7. The authors state that their transcriptional analysis reveals eight clusters of altered biological activity in HGPS, but it is not entirely clear what those eight specific clusters are based on the way the section is written.

Clarification of the specific clusters was added in the result section under “Gene ontology analysis of transcriptional changes reveals eight clusters of biological activity affected in HGPS”, in the second paragraph as follows:

“Overall, relevant gene ontology terms identified in the study pertain to eight functional clusters: DNA maintenance and Epigenetics (figure 1), Repair and Extracellular matrix (figure 2), Bone (figure 3), Adipose Tissue (figure 4), Blood Vessels (figure 5), and Muscle (figure 6)”

8. Figure 2C: why was the age-matched control comparison throughout this manuscript excluded from this particular analysis?

The age-matched control comparison was not excluded from this analysis. Instead, as shown in the heatmaps, there is no significant enrichment in these pathways in the age-matched controls. Rather, these effects are observed in comparisons with older control cohorts, which were the only ones included in the results table for clarity.

We have now clarified this in the figure legend as follows:

Figure 2

A) Heat map of RNA-seq transcriptome analysis for 585 selected genes related to repair. Data presented as in Figure 1B.

B) Heat map of RNA-seq transcriptome analysis for 145 selected genes related to extra cellular matrix. Data presented as in Figure 1B.

C) Summary table of processes related to repair and extra cellular matrix organization, represented as up or downregulation in transcription based on RNA-seq of young/teenager progeria patient derived fibroblasts, compared to middle age or old control patients. Age comparisons that yielded no significant results in relevant categories are not shown

9. The 10th paragraph of the Results, pertaining to epigenetic modifications, seems more suited to be presented earlier in the section where the authors present their GO analysis highlighting epigenetic programming and DNA maintenance (3rd paragraph of Results) with regards to Figure 1.

We apologize for the confusion, but this paragraph refers to a different analysis with further age stratification, as stated in the previous paragraph: “To further refine our findings related to young age patients and the mesenchymal phenotypes observed, we stratified the young HGPS patients into two age groups: early infancy (0-3 years old) and children (4-7 years old)”. We go on to describe (a) total number of genes that appear up and down-regulated, (b) the similarity of the HDAC GO terms in the 0-3 y/o cohort with the previous analysis, and finally (c) the differences observed in the 4–7-year-old cohort, where “ten percent are related to chondrogenesis events”, along with the genes identified.

10. 11th paragraph: The underlying theme of the manuscript is that defective mesenchymal stem cell commitment in HGPS results in abnormal development of bone. As such, the gene expression data related to ossification and calcium homeostasis should be provided. Furthermore, BMP4 stated to be upregulated, is not included in the heat plot associated with this finding (Figure 7A).

The bone-related genes that are found in the initial full comparison of all HGPS patients vs. controls are highlighted in Figure 3. This set actually includes BMP4, but there are too many genes in that figure to call out by name. Then, figure 7 shows additional endochondral ossification genes found specifically in the comparisons where Progeria patients were stratified by age. This includes WNT5a and many of its pathway targets, as highlighted in Figure 7a/b. We have now added a detailed examination of BMP4 expression in Figure 7c in order to highlight this comparison as well in this figure. We have also focused on the Hi-C data of BMP4 in Figure 9, adding to the coherence of the flow of the results.

Gene expression data related to ossification and calcium homeostasis (Zscore tables AND/OR gene counts tables, by patient, are also now provided in Figure 3, Source Data 1)

11. Downregulated genes are shown to be associated with alterations in compartment identity or lamin interactions. There is no mention if similar alterations occur for upregulated genes, which should be included.

The upregulated gene compartment comparison is shown in Figure 8F-H as well- the left panel of each set is upregulated and the right is downregulated. We agree that this was not clearly labeled in the figure and we have now added upregulated and downregulated headers over the plots for clarity as well as titles indicating which comparisons each panel shows. These graphs show that upregulated genes do not show a corresponding shift in compartment, so this is an effect specific to downregulated genes overall. We have now more clearly stated this in the text.

12. Some important findings in this study include the dysregulation of pathways involved in blood vessel morphogenesis and maturation, muscle development, and lipid metabolism. Vascularization is essential for normal bone development and tissue, muscle provides mechanical and metabolic influences on bone homeostasis, while lipid metabolism is critical in determining the phenotypic fate of common progenitors of osteoblasts and adipocytes. The authors should therefore comment on the potential roles dysregulation of these pathways has in the HGPS bone phenotype as an alternative to suggesting MSC depletion as the sole driving force behind HGPS pathology.

Our manuscript revisions attempt to clarify that we do not intend to claim that MSC depletion is the only driving force behind HGPS pathology. The reviewers are correct that each of these pathways and tissues can influence each other, and thus we have included statements that we observe gene misregulation across multiple mesenchymal lineage pathways.

We include a discussion of the potential relationship between vascular function and adipogenesis in the discussion paragraph which begins “Related to later differentiation events, the depletion of microvasculature…”

However, taking into account other reviewer recommendations, we also have revised the manuscript to ensure it is clear that everything we are measuring here is derived from what can be detected in fibroblasts. Fibroblasts can be a sentinel of mesenchymal lineage health in chronic disease and fibroblast samples from patients can include MSCs, therefore, we think speculating on the MSC pool is possible, but it would be overreaching to use fibroblast data to draw too many conclusions about vascularization and muscle mechanics in bone development.

We have changed the ideas and speculation section to address some of these concerns and to make the claims “softer”

Ideas and speculation

Overall, the cohorts of biological processes that we observe to be misregulated across different age groups of Progeria patients lead us to speculate the impact of this misregulation across a series of events in development, which will warrant further investigation:

A. Biology of priority: the formation of bone “fires” at specific times. These events are “non-negotiable” potentially taking precedence over other concomitant MSC fates. Deficits in endochondral ossification will condition the system to drawing from pericyte and tissue-resident MSCs in an attempt to compensate.

B. The loss of bone fusion and regression of bone length observed in teenaged HGPS patients is an indirect result of deficits in arrest of chondrogenesis proliferation, that occurred in early childhood. These deficits may result in depletion of MSC pools, both in the bone marrow and in tissues

C. Depletion of the pericyte niche is exacerbated by a chronic, incompetent, wound repair response. Ultimately, this depletion will result in loss of microvasculature integrity and subsequent vascular events observed in HGPS, such as vascular stiffness and atherosclerosis.

E. In adipose tissue, the activation of MSCs located at the pericyte niche for addressing deficient wound repair response or osteogenesis, could lead to a deficient differentiation into adipocytes

F. Elaborating on proposed MSC interventions (Infante and Rodriguez, 2021) and current gene editing efforts (Koblan et al., 2021), targeting the bone marrow niche in young patients could rescue the later HGPS phenotypes.

13. A major shortcoming of this work is the drawing of conclusions on pathomechanisms of HGPS in multiple mesenchyme-derived tissues based on fibroblast transcriptional and epigenetic profiles which are, however, acknowledged by the authors. Specifically, changes in gene expression are attributed to alterations in compartment identity or lamina association in fibroblasts, but no data is provided to suggest those same changes occur identically in other cells or tissues where conclusions are being made. This requires further discussion in the paper.

We have now specifically stated this as a limitation, along with the statement that other tissues of the mesenchymal lineage are not available from Progeria patients and a discussion of the evidence of how fibroblasts may report on the state of the mesenchymal lineage.

To investigate how some key genes we highlight behave during differentiation from MSC to osteocyte or adipocyte lineages, we analyzed data from studies that induced WT MSC differentiation into these lineages (GSE37558 and GSE151324). We found cases in which the patterns of gene expression in this differentiation trajectory could potentially inform an interpretation of misregulation in Progeria, such as shown with FZD4 (Author response image 1). This gene normally increases with osteoblast and adipocyte differentiation induction but stays aberrantly high in Progeria teens vs. WT teen and adult samples. However, in the process of this analysis, we identified a number of genes whose behavior was inconsistent between published datasets and thus likely sensitive to the source of MSCs and differentiation protocols. Further, taking into account the valid reviewer concerns that we do not want to appear to claim that we can comment on the Progeria MSC lineage directly, we decided it would add unhelpful confusion to include such data about comparisons to healthy MSC differentiation in the manuscript. Any conclusions we could draw from such comparisons would still be speculative, and therefore we have left further analysis in this direction to future work.

**Author response image 1. sa2fig1:** 

14. The manuscript would be strengthened by reorganizing the Results section to more clearly present the data pertaining to each of the five functional hubs that were found to be affected in HGPS cell lines in a linear manner.

In the revised manuscript we walk through 8 functional clusters that emerged from the comparisons of all HGPS children and teens to age-matched and adult controls: (1) DNA maintenance (2) Epigenetics (both discussed in Figure 1) (3) Tissue repair, (4) Extracellular matrix (Figure 2) (5) Bone (Figure 3) (6) Adipose tissue (Figure 4) (7) Blood vessel homeostasis (Figure 5) and (8) Muscle (Figure 6).

We then start a separate section to describe the results from a more detailed age stratification into infants and young children- stages where different phases of development are taking place. Here, we describe that for the youngest HGPS patients, epigenetic factors are most frequently misregulated, while for 4-7-year-old children, genes involved in ossification and chondrogenesis are most enriched in the misregulated set (Figure 7). We hope this separation of the sections clarifies the results.